# Hierarchical Goal Abstractions via Learned Subset Relations

Fabian Wurzberger [1]   Sebastian Gottwald [1]   Zeqiang Zhang [1]   Daniel Alexander Braun [1]

## Abstract

In self-supervised goal-conditioned reinforcement learning (RL) without external rewards, goals are typically specified by observations sampled from experience. However, depending on the observation structure, such a fixed representation of goals may be either too concrete (requiring exact pixel-level matches) or too abstract (involving ambiguous observations). Here we propose the construction of hierarchical latent goal spaces that integrate both concrete and abstract goals. To this end, we use an energy function to learn a partially ordered space, in which a subset relation between observations naturally induces a hierarchy from concrete to abstract goals. This representation enables agents to disambiguate specific states while also generalizing to shared concepts. In experiments on navigation and robotic manipulation, agents trained with our hierarchical goal space achieve higher task success and greater generalization to novel tasks compared to agents limited to purely observational goals.

## 1. Introduction

Over the last decade, reinforcement learning (RL) has achieved remarkable successes, both in mastering highly complex games and in other domains where environments provide clearly defined reward functions (Mnih et al., 2015; Silver et al., 2016; Vinyals et al., 2019; Hafner et al., 2025). In real-world applications, however, such reward functions are rarely available or are highly non-trivial to specify (Amodei et al., 2016; Russell, 2016; Christiano et al., 2017). In contrast, humans learn about the world through exploration, play, and observation, largely without external reward signals telling them what to do (Begus et al., 2014; Gopnik et al., 1999; Goupil et al., 2016). Another prob-

lem for real-world applications is task-specificity: an agent trained to solve one task may not be able to apply its learned skills to a different, even slightly modified task (Zhang et al., 2018; Delfosse et al., 2025). One approach to address both the lack of external rewards and the need for generalization is unsupervised RL, in which agents are pre-trained without relying on designed reward signals and later adapted to downstream tasks. This pre-training can enable faster inference of policies that generalize across a broad range of tasks. In this paper, we tackle the problem of hierarchical goal representations for unsupervised (pre-)training within the framework of goal-conditioned reinforcement learning.

**Unsupervised RL** is characterized by a spectrum of diverse techniques, including intrinsic motivation approaches based on novelty, learning progress, or empowerment (Salge et al., 2014; Zhang et al., 2021), latent skill learning (Eysenbach et al., 2019; Sharma et al., 2020), goal-conditioned RL with self-selected goals (Nair et al., 2018; Bae et al., 2025), approaches that approximate long-term dynamics with successor measures (Agarwal et al., 2025b;a), and contrastive RL methods (Laskin et al., 2020; Schneider et al., 2021; Eysenbach et al., 2022).

**Goal-conditioned RL** (Kaelbling, 1993) extends classical RL by conditioning rewards, policies, and value functions on goals, that effectively serve as an index for multiple tasks that require different desired behaviors. Here, we focus on goal-conditioned RL within a self-supervised learning setting where tasks are specified by the goals the agent attempts to achieve without providing explicit external rewards. This raises the question of how to represent goals alternatively to rewards. A prominent example approach is contrastive reinforcement learning (Eysenbach et al., 2022), where hindsight learning (Andrychowicz et al., 2017) is harnessed to relate current states and actions to future outcomes that are treated as goals. This is achieved by contrastively learning a value function that has high values for matching pairs of state, actions, and outcomes and low values for random pairings.

**Goal representations** in the literature are often expressed directly in observation space (Ghosh et al., 2021; Eysenbach et al., 2022), which ties the agent's capabilities to both the abstraction level and the structure of the underlying observation space. Typically, environments either provide local observations like egocentric images, or global observations

[1]Institute of Neural Information Processing, Ulm University, Ulm, Germany. Correspondence to: Fabian Wurzberger <fabian.wurzberger@uni-ulm.de>.

*Proceedings of the 43rd International Conference on Machine Learning*, Seoul, South Korea. PMLR 306, 2026. Copyright 2026 by the author(s).

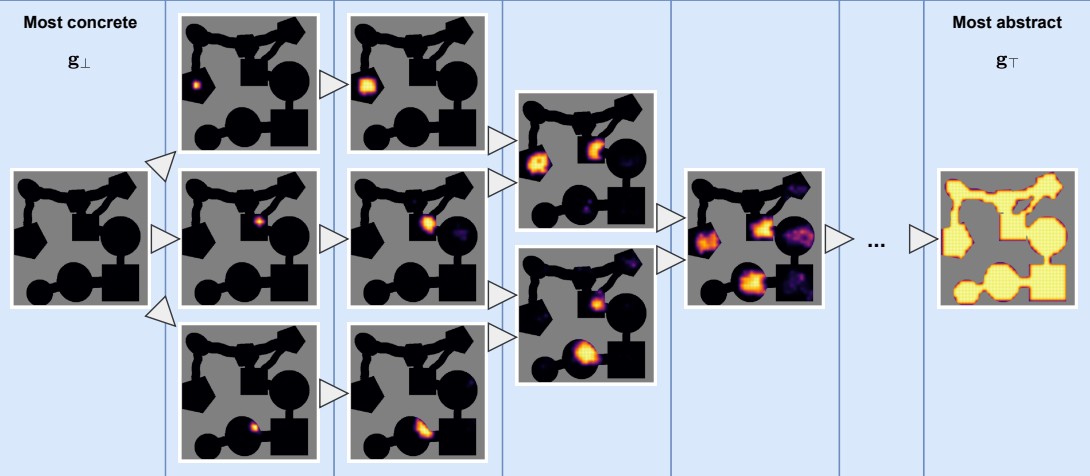

*Figure 1.* Traversing the latent goal space. Directed edges denote the partial order, showing that abstract goals can be formed by either expanding the coverage of concrete goals or merging multiple concrete goals into more abstract goals.

such as precise states and positions. These modalities inherently bias how goals are interpreted: local observations may be ambiguous, leading to abstract goals unless additional context is provided, while global observations specify concrete states but make it harder to capture higher-level abstractions through composition. Thus, neither local nor global observations alone provide a sufficient basis for representing goals, as each is biased towards a single level of abstraction. While the most concrete goals can be seen as elements of a sample space composed of observations or sequences of observations, more abstract goals can instead be seen as (soft) partitions on this space. In general, such goals, that consist of multiple observations, can be induced by constraints (Colas et al., 2019), by utility functions (Christiano et al., 2017; Vamplew et al., 2024) or by desired distributions over observational states (Pong et al., 2020; Ziebart et al., 2008). However, if different levels of abstraction are defined by separate functions or constraints (Sutton et al., 1999; Ho et al., 2019), then it might be difficult to capture their relationship to lower-level goals. An alternative is to represent both concrete and abstract goals in a shared latent space, analogous to word embeddings, where both tokens and longer texts can be encoded in the same vector space (Mikolov et al., 2013).

In the following, we thus propose to study hierarchical latent goal spaces with superimposed partial orders, where all goals are represented as vectors in a shared space organized by varying levels of abstraction (see Figure 1). This structure supports join and meet operations that make goals more or less abstract within the hierarchy. In particular, the partial order is learned via an asymmetric energy function (LeCun et al., 2006; Grathwohl et al., 2020) that indicates whether one goal forms a subset of another (i.e. is a more concrete instance). We validate this approach in the context of spatial

abstraction, where goals encode sets of observations related by temporal proximity, using contrastive training to learn the containment relationships.

The paper is organized as follows. In Section 2 we introduce our methods, including the latent space encoding, the energy function training and the evaluation methods. For evaluation after pre-training, we confront agents with multiple novel reward functions and we search for the best fitting goal in the trained latent space to represent these reward functions. In Section 3, we illustrate our method in three different RL environments (GridWorld with LiDAR, MemoryMaze, FetchPush). We first illustrate the best fitting goals to represent abstract reward functions, and then show the performance of the corresponding pre-trained goal-conditioned policies when searching for the best-fitting goal. In Section 4 we discuss our approach in the context of the wider literature and conclude in Section 5.

## 2. Methods

**Hierarchical goal spaces.** Abstraction can be naturally understood as a subset relation: a concrete goal (e.g. this corner) may be considered as subset of a set of concrete goals (e.g. blue corners), which may in turn be a subset of a larger set of subsets (e.g. all corners), where each subset shares some similarity. Of course, there may also be sets of goals that share an intersection, but are not contained within each other (e.g. corners and blue things). Mathematically, the subset relation is a partial order, because it is reflexive, antisymmetric, and transitive. The key idea is to arrange goals according to this partial order, forming a lattice that runs from the most specific to the most general goal. We formalize this idea through a shared latent goal space, $\mathcal{G}$, which supports both representing goals at varying

degrees of abstraction and traversing between abstraction levels. Formally, we capture different levels of abstraction by imposing a partial order $\preceq$ on the latent goal space: If $\mathbf{g}_a \preceq \mathbf{g}_b$, then $\mathbf{g}_b$ represents a more abstract goal than $\mathbf{g}_a$. The most concrete goals are denoted by $\mathbf{g}_\perp$, and the most abstract by $\mathbf{g}_\top$, such that

$$\mathbf{g}_\perp \preceq \mathbf{g} \preceq \mathbf{g}_\top \quad \forall \mathbf{g} \in \mathcal{G}.$$

Using the lattice structure of this partial order enables the traversal of the latent space through join, $\mathbf{g}_a \vee \mathbf{g}_b = \inf\{\mathbf{g} \mid \mathbf{g}_a \preceq \mathbf{g}, \mathbf{g}_b \preceq \mathbf{g}\}$ and meet operations, $\mathbf{g}_a \wedge \mathbf{g}_b = \sup\{\mathbf{g} \mid \mathbf{g} \preceq \mathbf{g}_a, \mathbf{g} \preceq \mathbf{g}_b\}$. Intuitively, the join identifies the least abstract goal that encompasses both $\mathbf{g}_a$ and $\mathbf{g}_b$, while the meet identifies the most concrete goal they share.

**Energy function and subset relation.** Similar to how total orders can be represented as unary utility functions under certain conditions (Von Neumann & Morgenstern, 1947; Debreu, 1954), there are multiple equivalent ways to represent partial orders, such as multi-utilities (Evren & Ok, 2011), injective monotones (Hack et al., 2022), or the characteristic function $\chi_\preceq$ of the partial order directly. Specifically, we can model the latter as an "energy" $E_\theta(\mathbf{x}, \mathbf{y})$ that estimates whether $\mathbf{x} \preceq \mathbf{y}$, i.e.

$$E_\theta(\mathbf{x}, \mathbf{y}) \approx \chi_\preceq(\mathbf{x}, \mathbf{y}) = \begin{cases} 1 & \mathbf{x} \preceq \mathbf{y} \\ 0 & \text{else.} \end{cases}$$

While this type of representation does not inherently enforce order properties such as transitivity, it is sufficiently general to learn any relation given enough data. In practice, however, we do not expect to perfectly approximate the characteristic function. Instead, our approach views the order structure as an inductive bias, yielding a hierarchically organized latent space while retaining flexibility to deviate from a strict order. Approximating the characteristic function, however, offers a clear operational advantage: it enables traversal of the induced hierarchy through optimization, holding one input fixed while updating the other. More precisely, we can move upward to find a more abstract goal ($\uparrow$) by maximizing $E_\theta$ over the second entry while keeping the first fixed, or move downward to obtain a more concrete goal ($\downarrow$) by maximizing over the first entry while keeping the second fixed. In particular, given a set of goals $\{\mathbf{g}_i\}_{i=1}^n$, we can find a new goal that is either more abstract or more concrete than the entire set through gradient ascent:

$$\mathbf{g}_\uparrow^{(t+1)} = \mathbf{g}_\uparrow^t + \eta \nabla_{\mathbf{g}_\uparrow}\Big|_{\mathbf{g}_\uparrow = \mathbf{g}_\uparrow^t} \sum_{i=1}^n E_\theta(\mathbf{g}_i, \mathbf{g}_\uparrow)$$
$$\approx \mathbf{g}_1 \vee \cdots \vee \mathbf{g}_n \quad \textit{(more abstract)}$$
$$\mathbf{g}_\downarrow^{(t+1)} = \mathbf{g}_\downarrow^t + \eta \nabla_{\mathbf{g}_\downarrow}\Big|_{\mathbf{g}_\downarrow = \mathbf{g}_\downarrow^t} \sum_{i=1}^n E_\theta(\mathbf{g}_\downarrow, \mathbf{g}_i)$$
$$\approx \mathbf{g}_1 \wedge \cdots \wedge \mathbf{g}_n \quad \textit{(more concrete)}$$

(1)

where $\mathbf{g}_\downarrow^0, \mathbf{g}_\uparrow^0 \sim \mathcal{G}$ are initialized randomly to ensure diversity of optimized goals. In Appendix B.5, we test the ability of our learned energy model to capture the transitivity of goal sequences generated by Equation 1.

**Spatial abstraction.** In a self-supervised setting, the data available to the agent consists solely of action–observation sequences, lacking external rewards or predefined similarity measures in observation-action space. To discover useful abstractions, the agent can group similar observations according to their temporal proximity. A large set of observations that contains a smaller set as a subset can then be regarded as a "spatial" abstraction that covers a broader range of possible observations. In the following, we use observation sequences as a means to construct such sets of spatially related observations. We distinguish goals based on their spatial coverage: longer trajectories covering broader areas are encoded as more abstract goals, while shorter trajectories covering less space are treated as more concrete. Thus, we use the subset relation between sequences as a proxy for this spatial hierarchy, where a sequence contained within a larger one corresponds to a more concrete goal. In a separate ablation study, we confirm that these observation sequences $\tau$ serve as suitable proxies for observation sets, independent of temporal order (cf. B.2).

In practice, we employ a single contrastive learning framework to jointly learn the trajectory encoding $\phi_\theta(\tau)$ and the asymmetric energy $E_\theta(\phi_\theta(\tau), \phi_\theta(\tau'))$ end-to-end (see Figure 2). The energy function is optimized to output close to 1 if $\tau$ is a subsequence of $\tau' = (\mathbf{o}_0, \ldots, \mathbf{o}_T)$ and close to 0 otherwise, guided solely by temporal proximity. To handle sequences of arbitrary length, we use a recurrent neural network (RNN)-based encoder (Cho et al., 2014), which maps the sequence $\tau$ into a fixed-dimensional context vector $\mathbf{c}_\tau$. Similar to Hafner et al. (2022), we further encode this context vector into a discrete latent representation $\mathbf{g}$ using a categorical encoder, where the discrete bottleneck encourages the formation of higher-level abstractions over goals. Differentiability is maintained by using a straight-through estimation of the gradients with respect to the discrete samples. This setup is closely related to the discrete variational autoencoder (VAE) used by Hafner et al. (2025), but instead of optimizing a reconstruction loss, we train the model with a reconstruction-free objective based on our proposed subset relation, using contrastive learning.

Given recorded data $\mathcal{D} = \{\tau_i\}_i$ consisting of trajectories $\tau_i = (\mathbf{o}_0, \ldots, \mathbf{o}_T)$, we construct datasets $\mathcal{D}_+$ and $\mathcal{D}_-$ of positive and negative pairs of trajectories, respectively, based on their relation in sequence space. The energy function $E_\theta$ and sequence encoder $\phi_\theta$ are learned jointly by optimizing

$$\mathcal{L}(\theta) = -\mathbb{E}_{(\tau_1, \tau_2) \sim \mathcal{D}_+}\big[\log E_\theta(\phi_\theta(\tau_1), \phi_\theta(\tau_2))\big] \quad (2)$$
$$-\mathbb{E}_{(\tau_1, \tau_2) \sim \mathcal{D}_-}\big[\log(1 - E_\theta(\phi_\theta(\tau_1), \phi_\theta(\tau_2)))\big].$$

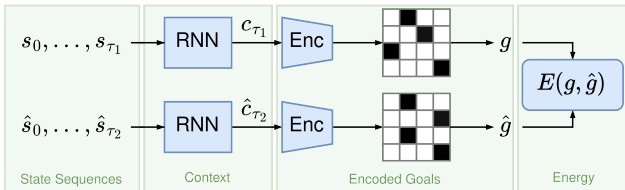

*Figure 2.* Proposed sequence abstraction model combining a learnable goal representation and similarity measure.

Here, $\mathcal{D}_+$ contains pairs $(\tau_1, \tau_2)$ such that $\tau_1 \subset \tau_2$, including cases where both come from the same trajectory as well as composite trajectories formed from unrelated segments. In contrast, $\mathcal{D}_-$ contains pairs where $\tau_1 \not\subset \tau_2$, for example pairs $(\tau_1, \tau_2)$ with $\tau_2 \subset \tau_1$, as well as non-overlapping trajectories, either from the same or from distinct base trajectories $\tau$. Examples are shown in Figure 3. The complete algorithm is detailed in Algorithm 1.

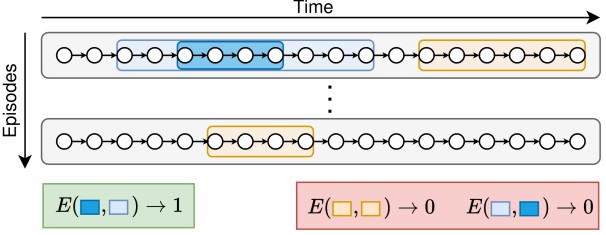

*Figure 3.* Subset selection for contrastive learning. Positive (green) and negative (red) examples are selected from observation trajectories.

**Abstract hindsight relabeling.** To learn policies in our approach, we choose contrastive reinforcement learning (CRL) which is based on hindsight relabeling of achieved goals (Eysenbach et al., 2022). In standard CRL, relabeling is restricted to single observations from past experience. This limits the diversity of goals available for training, since the agent repeatedly encounters only a narrow subset of the goal space, an effect that becomes more pronounced when training on a fixed, concrete task. Our goal encoding addresses this limitation by enabling the generation of more abstract goals from a concrete goal. This provides the agent with richer supervision: it can learn from both fine-grained, single-observation goals and higher-level, temporally extended goals. Such abstract goals can be obtained by (i) directly encoding sequences of observations instead of single observations, (ii) optimizing for a more abstract goal by using Equation 1 (*energy-driven*), or (iii) finding compositional goals based on rewards optimizing Equation 3 (*reward-driven*), as formalized in the next paragraph. In practice, we randomly sample the sequence length for relabeling using approach (i), allowing the agent to experience both more concrete and abstract goals. For (iii), a suitable

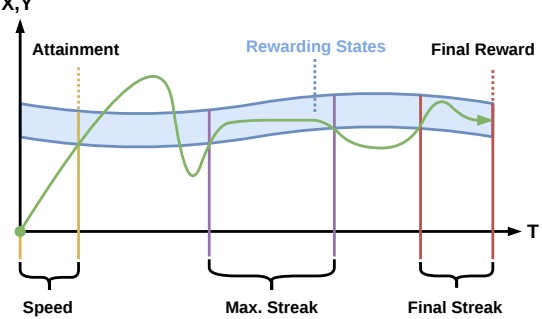

| Criterion | Description |
|---|---|
| Attainment | Rewarding state reached |
| Total Reward | Time fraction at reward |
| Final Reward | Reward in the final state |
| Final Streak | End consecutive rewards |
| Max. Streak | Max consecutive rewards |
| Avg. Streak | Avg. consecutive rewards |
| Speed | Steps to first reward |
| Alignment | Reward-goal similarity |

*Figure 4.* Agent performance criteria.

intrinsic reward is compatibility based on gradient magnitude, merging latent goals that are easy to combine into compositional goals. We hypothesize that a broader and more expressive set of goals improves generalization and task performance. This is supported by an ablation study that can be found in B.4, where dropping the optimization step for further abstraction using (ii) or (iii) causes a substantial decline in performance. In the following experiments, we use (iii) to train our agents. For a fair comparison, we use the same total number of goals for training each algorithm.

**Generalization to novel rewards.** We draw inspiration from zero-shot RL, evaluating agents on novel, unseen reward functions to measure their *persuadability* (Levin, 2022). To compare different goal representations, we first pre-train each agent on the same single-observation goal-reaching task using its respective goal space. After pre-training, each agent explores the environment autonomously to translate previously unseen reward functions into corresponding goals. During this phase, the agent performs fixed-length rollouts and collects rewards while acting under the policy induced by its current best-fitting goal. Importantly, this goal is not static: it is continuously optimized based on the agent's observations and the reward feedback it receives. This setup allows us to assess how effectively each goal representation supports adapting to new reward functions as well as how agents behave under the new goals.

In practice, we employ a hindsight learning procedure that translates rewarding observations into goals. To this end, we minimize the binary cross entropy between rewards and

similarities of parametrized $\mathbf{g}_\vartheta$ and achieved goals $\mathbf{g}_t$. More precisely, optimal goals $\mathbf{g}_{\vartheta^*}$ are obtained through

$$\vartheta^* = \arg\min_{\vartheta} \; \mathbb{E}_{(r_t, \mathbf{g}_t) \sim \mathcal{D}} \Big[ \mathrm{BCE}\big(r_t, \mathrm{Sim}(\mathbf{g}_t, \mathbf{g}_\vartheta)\big) \Big], \quad (3)$$

where $\mathcal{D}$ are reward-goal tuples obtained from environment interactions and $\mathrm{Sim}$ denotes any similarity measure between goals. We evaluate the agent's performance given the criteria in Figure 4 that capture different aspects of the learned goal space and policy, such as goal attainment and reward alignment. We define the alignment score as the cosine similarity between the ground-truth rewards and the similarities computed between the optimized goal and a fixed set of uniformly distributed observations.

## 3. Experiments

We evaluate our approach on different environments with local and global observations. For our approach, observations are encoded into goals using the sequence encoder, and the similarity measure is our energy function, $Sim = E_\theta$. We compare our method against dual goal representations (DGR) (Park et al., 2026), which utilizes a bilinear similarity measure $Sim(s, g) = \psi(s)^T g$. We also include a comparison to purely observational goals ($g = o$) with cosine similarity $Sim(g, g') \propto g^T g'$. We focus our main analysis on observational goals, as this is the standard approach, and DGR, which represents a strong baseline on various goal-conditioned tasks (Park et al., 2026). Results for additional baselines, including temporal representation alignment (TRA) (Myers et al., 2026), value implicit pre-training (VIP) (Ma et al., 2023), and a $\beta$-VAE (Higgins et al., 2017; Nair et al., 2018), are detailed in Appendix B.

Our experiments consist of two phases:

1. **Pre-training:** Policies and goal representations are pre-trained on the original goal reaching task with single observational goals provided by the environment. Observations are encoded into the corresponding goal space, and we jointly train the policy and goal representation while the agent interacts with the environment.

2. **Evaluation:** We evaluate the frozen pre-trained policies and goal representations on novel reward functions. We find best matching goals to represent the reward functions by optimizing Equation 3. For our approach, we learn a representation of the reward function at hand by optimizing for a more abstract goal than all rewarding goals. During this phase, the agent has to discover the reward function only through environment interactions.

**Goal visualization.** In order to get an intuition of what the goal space actually encodes, we use heatmaps to visualize a

particular target goal as follows. First, we sample a set of diverse observations covering possible positions in the environment. Next, we encode them as single-observation goals and compare them to the target goal based on the respective similarity measure. Then we create a heatmap by using the position information and the computed similarities.

### 3.1. Environments

We use two navigation tasks with ego-centric, local observations. For post-training analysis, we define observation-based rewards (e.g. circular room, yellow corner) as well as spatial rewards (specific room, disjoint rooms). We also use a robot manipulation task with a global, image-based observation space with gripper and object-based reward functions. Rewards are binary and state-based. A full list of rewards is given in A.3.

**GridWorld** is a 2D navigation task where agents navigate through a grid-based environment using discrete actions. A new episode starts when the agent reaches the goal. Contrary to normal position-based GridWorld environments, our agent receives egocentric LiDAR observations.

**MemoryMaze** (Pasukonis et al., 2022) is originally a long-term memory task based on discrete 2D navigation with image observations. We adapt the task to use the environment for goal-based navigation where goals are provided as image observations. Episodes end when the goal is reached. **FetchPush** (Plappert et al., 2018) is a robotic manipulation task where a gripper is tasked to move an object to a specific position and keep it there until the episode ends. We use the adapted task from Eysenbach et al. (2022) to obtain image-based observations, making the environment challenging. Goal images show the object at some random position with the gripper close to the object. We define novel gripper rewards (e.g. moving the gripper to the table border) and object rewards (e.g. pushing the object off the table), where the critical question is whether goals can represent both.

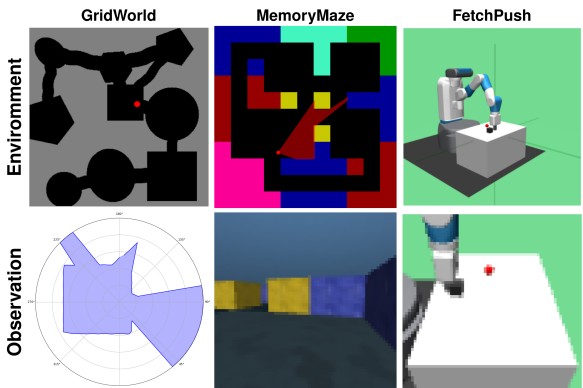

*Figure 5.* Evaluated goal-reaching environments and their observations, ranging from navigational tasks to robotic control.

## 3.2. Results

**Traversing the hierarchy.** Optimizing the energy function to traverse the induced hierarchy proves effective for generating goals at varying levels of abstraction. Figure 1 shows how we can traverse the hierarchy in the GridWorld environment. Starting from the most concrete goal, we can optimize for single-position goals, abstract them to broader regions and even combine individual regions to more abstract, disjoint regions. Finally, the hierarchy is bounded by the most abstract goal. Note that only one possible abstraction is shown at a time, while the partial order allows for different abstractions to fulfill the subset relation. With this observation in mind, we show how we can learn abstract representations of rewards by combining individual goals, corresponding to high rewarding states, into a single abstract goal.

**Representing reward functions with goals.** Our goal representation demonstrates the ability to encode diverse reward functions in the GridWorld environment, as shown in Figure 6. These range from concrete spatial targets (specific rooms) to increasingly abstract concepts (room centers, disjoint rooms). This spectrum reveals critical differences in representational requirements: while some rewards correspond directly to observable features (e.g. passages that can be identified from single observations), others demand spatial and compositional knowledge of the environment. Our analysis reveals limitations with respect to observational goals. While raw observations can partially encode simple, visually-identifiable objectives like corridor passages, they fail to capture spatial relationships and compositional properties effectively. Furthermore, the reward-optimizing observations show overall high and noisy activity in cosine similarity, indicating fundamental limitations in their capacity to serve as robust abstract goal representations. In contrast, DGR proves highly effective at capturing the reward functions, achieving results comparable to our approach. We attribute this to its bilinear compatibility function and unbounded latent space. More results supporting our interpretation in the MemoryMaze and FetchPush environments are given in B.

**Policy adaptation to novel rewards.** Figure 7 presents a comparative analysis of agents' policies when exposed to the goals that best represent the novel reward functions according to Equation 3. We evaluate the policies according to the performance criteria outlined in Figure 4. Our findings reveal comparable performance of the different agent types in tasks where the available observations are able to capture meaningful aspects of the reward functions. However, a difference emerges for spatial tasks that demand both spatial and compositional knowledge about the environment. Our goal representation demonstrates better performance

across most evaluation metrics for spatial rewards, while agents using observational goals exhibit near-random performance. This performance gap indicates that agents trained with our goal representation have better generalization capabilities for abstract goals that cannot be adequately represented within the observation space. DGR shows similar task performance to our approach on these spatial tasks, while maintaining an overall higher goal alignment score on all tasks.

In the FetchPush environment, where all observations are global, we analyze the agents' behavior on abstract gripper and object-based rewards. We deliberately excluded concrete spatial objectives (such as positioning objects or grippers at specific coordinates) due to their extremely low success probability without sufficient exploration. Our results demonstrate that our goal representation successfully handles both abstract object manipulation and gripper control tasks, performing considerably better than the DGR-trained policy on object-based and gripper-based rewards. In contrast, observational goals fail on both tasks. The high reward-alignment scores of DGR, paired with its decreased goal attainment, indicate that while the bilinear compatibility measure is able to capture the reward structure, there is no trivial mechanism to traverse the hierarchy and thus effectively train the agent on these abstract goals.

## 4. Related Work

**Self-supervised representation learning.** Self-supervised representation learning aims to learn meaningful representations without explicit supervision signals like labels (Ericsson et al., 2021). Contrastive learning has emerged as a powerful paradigm for self-supervised representation learning by augmenting data with labels, often obtained by defining a suitable similarity measure between data points (Jaiswal et al., 2021). The core principle involves bringing semantically similar samples closer together while pushing dissimilar samples apart in the learned embedding space. This approach has shown remarkable success across domains, including computer vision with methods like SimCLR (Chen et al., 2020) as well as natural language processing through approaches like contrastive sentence representation learning (Kim et al., 2021) and supervised contrastive learning for pre-trained language model fine-tuning (Gunel et al., 2021). Temporal contrastive learning extends these principles to sequential data by leveraging temporal relationships as augmentations. Contrastive learning through time (Schneider et al., 2021) takes inspiration from biology to learn object representations by forming augmentations from successive views in temporal sequences. Contrastive learning can be understood through the lens of energy-based models, where the similarity measures used to bring positive pairs

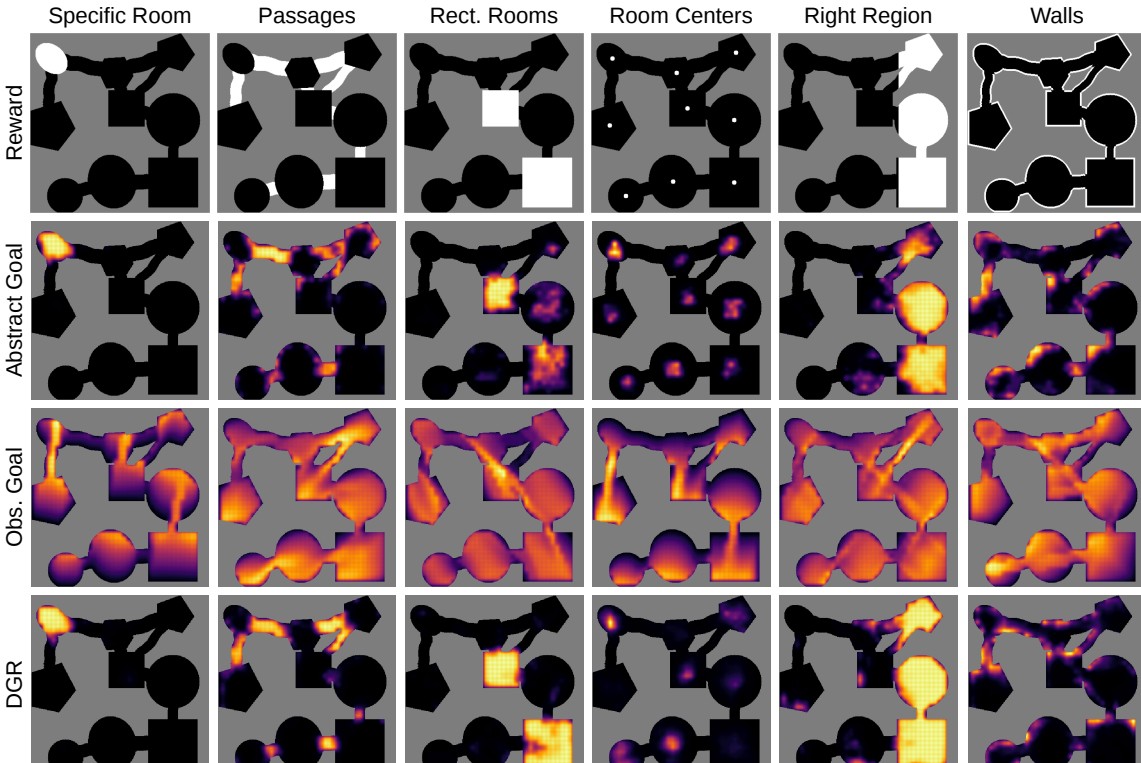

*Figure 6.* Encoded reward functions in the GridWorld environment. White regions in the top row indicate ground truth rewarding states. Notably, a single abstract goal can represent each reward function, whereas no single observation is sufficient.

together and push negative pairs apart implicitly form an energy landscape over the representation space (LeCun et al., 2006). Such energy functions can be used to learn compositional concepts as shown in Du et al. (2021). Our approach combines the idea of using temporal contrastive learning in combination with an energy function to guide representation learning. Moreover, we extend temporal contrastive learning beyond simple relations between individual images. By introducing a subset similarity in sequence space, we exploit temporal relations between entire trajectories to structure the latent representation.

**Abstraction in RL.** The RL literature mainly distinguishes state abstraction and temporal abstraction (Abel, 2020). While temporal abstraction is usually studied in some form of options framework (Sutton et al., 1999), state abstraction reduces the size of the state space by engineering or learning low-dimensional features from raw sensory inputs (Mnih et al., 2015). Both kinds of abstraction can be conceptualized in terms of subsets, where state abstraction can be understood as a subset structure on the set of states and temporal abstraction as a subset structure on the set of trajectories. Often, state abstraction is achieved with reconstruction-based compression methods like VAEs that do not always focus on relevant features (Ha & Schmid-

huber, 2018; Hafner et al., 2019; 2025). Hence, there has been a flurry of reconstruction-free compression methods based on contrastive learning (InfoNCE and contrastive predictive coding (Oord et al., 2018; Ma & Collins, 2018), or DeepInfoMax (Hjelm et al., 2019)). Our proposed method for goal embeddings with spatial abstraction follows this line of research but adds the partial-order structure on the latent space that is absent in previous methods. Observation trajectories also exhibit some level of temporal abstraction that we did not explore in the current study. In the future, it will be interesting to pursue this avenue in the context of hierarchical RL (Vezhnevets et al., 2017; Nachum et al., 2018) where concrete and abstract goal vectors could be used to communicate between different agents in the hierarchy. Unsupervised pre-training goal-conditioned policies with such goals could also form the basis to discover a diverse set of skills (Eysenbach et al., 2019).

**Representation learning in self-supervised RL.** The concepts of goals, skills, and intentions share fundamental similarities as they all represent desired outcomes or behaviors that guide agent decision-making. Ghosh et al. (2023) learned intention-conditioned value functions by encoding how outcome likelihoods change when the policy acts with a particular intention in mind. Skill-based methods

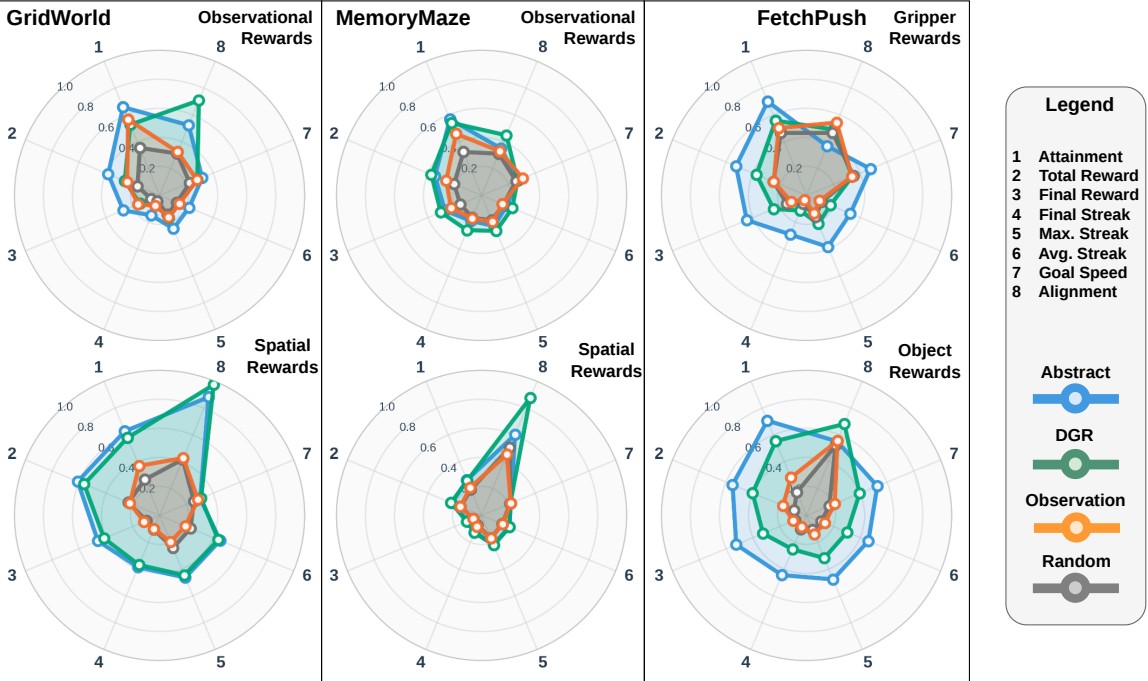

*Figure 7.* Performance on novel reward functions. Star plots show the normalized performance on various metrics (larger area is better). Our approach maintains robust performance across all domains, specifically excelling in solving spatial and object-manipulation tasks where observational goals fail to capture the necessary abstract structure and DGR lacks in goal attainment.

aim to find latent representations of reproducible behavior (Eysenbach et al., 2019; Sharma et al., 2020). To find latent skills, information theoretic ideas are employed to maximize mutual information between states and skills while making skills distinguishable (Eysenbach et al., 2019), between skills and future outcomes (Sharma et al., 2020) or between skills and state transitions (Laskin et al., 2022). Another way to encode diverse behavior is to exploit the linear dependence of the $Q$ value function on the reward (Touati & Ollivier, 2021; Agarwal et al., 2025b). Agarwal et al. (2025b) show that any agent behavior which can be represented by visitation distributions can be described as an affine combination of policy-independent basis functions. While our current embedding space is based on observational sequences and thus more related to goals, extending this approach to sequences of actions or action and observation sequences may support a representation that forms a bridge between goals and skills.

**Goal-conditioned RL.** Goal-conditioned RL enables agents to learn diverse policies by conditioning behavior on desired goals. As the overall objective is attaining a goal, rewards are in general sparse which poses a fundamental problem. Hindsight experience replay (Andrychowicz et al., 2017) and in general hindsight relabeling proved to be an effective method to tackle sparsity by relabeling failed trajectories as success under a different goal (Andrychowicz

et al., 2017; Ghosh et al., 2021). Different extensions to this idea were proposed to deal with dynamic goals (Fang et al., 2019) or prioritize the experience for better relabeling (Zhao & Tresp, 2018). Ghosh et al. (2021) rephrased the goal-conditioned learning problem as supervised learning without rewards, by relabeling experienced trajectories as success and using self-imitation learning to directly optimize the policy. Eysenbach et al. (2022) demonstrated that contrastive learning can be reinterpreted as goal-conditioned RL, showing that contrastive objectives naturally lead to goal-reaching behavior. Our work focuses on the representation of the goal space, which is orthogonal to the specific policy optimization method. Although we focus on contrastive RL (Eysenbach et al., 2022) to train agents in our experiments, our hierarchical representation can generally be used by other goal-conditioned RL algorithms.

**Goal representation learning.** While observations are commonly used as goals, various approaches have been proposed for learning latent goal representations. Nair et al. (2018) employed a VAE with reconstruction loss to embed observations into a latent space, enabling the sampling of novel goals and computation of distances in latent space. Building on this foundation, Nair et al. (2020) extended the approach using a conditional VAE that incorporates future goals to encode goal feasibility. Co-Reyes et al. (2018) took a different approach by encoding entire trajectories

into latent representations using a VAE trained with reconstruction loss, subsequently training a policy conditioned on these latent trajectories to replicate the encoded sequences. Hafner et al. (2022) utilized discrete latent sub-goals derived from a discrete VAE applied to world model states to guide reward optimization. We build on both ideas, encoding trajectories into latent representations and using a discrete VAE for encoding goals but in contrast to other approaches, we use reconstruction only to facilitate initial convergence before fully transitioning to contrastive learning. Ma et al. (2023) uses temporal contrastive learning to learn a goal representation based on the distance between temporally related observations. In contrast to our representation, the distance function used by Ma et al. (2023) is symmetric and thus does not encode a partial order. Park et al. (2026) and Myers et al. (2026) encode goals by using a bilinear similarity function trained with contrastive learning. While these approaches also learn an asymmetric compatibility measure, observations and goals are encoded into different latent spaces, making traversal of the induced latent space non-trivial. As demonstrated in our experiments, this hinders the agent's ability to effectively optimize policies for highly abstract goals, despite the bilinear representation's capacity to encode the reward structure itself.

## 5. Conclusion

In this paper, we introduced a novel approach for representing goals as embedding vectors in a hierarchical latent space with varying levels of abstraction. Existing methods typically define goals either through hand-engineered, goal-dependent reward functions or directly in terms of observations, thereby constraining the level of abstraction to the properties of the observation space. We demonstrated the feasibility of unified goal embeddings for spatial abstraction by encoding sequences of observations into latent goals via unsupervised contrastive learning. Crucially, the learned partial order naturally induces a hierarchy ranging from spatially focused concrete goals to broad abstract goals. By traversing this hierarchy to reach previously unseen goals, we can effectively encode novel reward functions within the latent space. Through experiments in navigation and robotic manipulation, we have demonstrated that agents trained with our hierarchical goal space achieve higher task success and significantly greater generalization to novel, unseen tasks compared to agents reliant on purely observational goals. Furthermore, while alternative representations like DGR can successfully encode abstract reward structures, our approach provides a critical mechanism for traversing the hierarchy. By enabling hierarchy traversal, our method bridges the gap between simply capturing abstract goals and actually using them to train adaptable policies.

Although our goal representation has been shown to be ef-

fective in encoding a variety of reward functions, several challenges remain. First, we focused on a single type of abstraction; however, many other forms are conceivable, such as those based on feature similarity across trajectories. Second, a distribution shift exists during pre-training: the agent's policy currently encounters mostly locally abstract goals rather than the highly abstract ones the model can represent, even when using our proposed hierarchy traversal approaches. Consequently, there is a discrepancy between the full expressive capacity of the goal representation and what the pre-trained agent can practically achieve. Future work should therefore focus on developing effective mechanisms, such as curriculum learning, to enable agents to fully utilize these rich representations. Furthermore, compositionality and multi-level abstraction warrant further exploration, particularly in the context of hierarchical RL, where goal vectors could facilitate efficient communication between different modules. Finally, we note that our method introduces computational overhead by encoding sets of observations as goals instead of single observations and requires additional optimization in latent space for our abstract hindsight relabeling approach (cf. Appendix C).

Ultimately, our approach can be integrated with various RL methods to foster unsupervised learning and generalization in the absence of explicit rewards, marking a critical step toward applying RL to open-ended environments.

## Impact Statement

This paper presents work aimed at advancing the field of self-supervised RL. By introducing a method to learn hierarchical goal representations without explicit reward engineering, our work reduces the dependency on human supervision and improves the ability of agents to generalize to novel tasks. This has potential applications in developing more autonomous robotic systems capable of operating in unstructured environments. We do not foresee immediate negative societal consequences beyond the general considerations associated with the development of autonomous decision-making systems.

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

# A. Implementation Details

## A.1. Code and Reproducibility

We provide a reference implementation on GitHub[1] to illustrate the core concepts of our approach on the GridWorld environment.

## A.2. Trajectory Selection for Contrastive Learning

For learning our representation using contrastive learning as described in Section 2 we first sample trajectories of length $T \sim \text{Unif}(3, 50)$ from the experience. We then independently sample sub-trajectories of length $\text{Unif}(1, \lfloor T/3 \rfloor)$. This formulation guarantees that the original trajectories of length $T$ can accommodate two non-overlapping sub-trajectories, each with a maximum length of $\lfloor T/3 \rfloor$, separated by a gap of at most $\lfloor T/3 \rfloor$ steps. The pseudo-code below details the trajectory selection process, in which $\mathcal{D}_+$ and $\mathcal{D}_-$ serve as the positive and negative sets for contrastive learning.

---

**Algorithm 1** Construction of $\mathcal{D}_+$ and $\mathcal{D}_-$

---

1: **Input:** Set $\mathcal{D}$ of observation trajectories, batch size $N$
2: **Output:** $\mathcal{D}_+, \mathcal{D}_-$
3: Let $\mathcal{D}_+ = \emptyset, \mathcal{D}_- = \emptyset$
4: **for** $N$ times **do**
5:      Sample $\tau_1, \tau_2, \tau_3 \sim \mathcal{D}$ s.t. $\tau_1 \subset \tau_2$
6:      $\mathcal{D}_+ = \mathcal{D}_+ \cup \underbrace{\{(\tau_1, \tau_2)\}}_{\text{Sub-trajectories}} \cup \underbrace{\{(\tau_1, \tau_1 \oplus \tau_3)\}}_{\text{Composite}} \cup \underbrace{\{(\{\mathbf{o}\}, \tau_1) \mid \mathbf{o} \in \tau_1\}}_{\text{Individual states}}$
7:      where $\oplus$ denotes concatenation along time.
8: **end for**

9:
10: **for** $N$ times **do**
11:      Sample $\tau_1, \tau_2 \sim \mathcal{D}$ and $\tau, \tau' \subset \tau_1$, s.t. $\tau \cap \tau' = \emptyset$
12:      $\mathcal{D}_- = \mathcal{D}_- \cup \underbrace{\{(\tau_1, \tau_2)\}}_{\text{Different traj.}} \cup \underbrace{\{(\tau, \tau')\}}_{\text{Disjoint segments}} \cup \underbrace{\{(\tau_4, \tau_3) \mid (\tau_3, \tau_4) \in \mathcal{D}_+\}}_{\text{Inversion}}$
13: **end for**

---

## A.3. Environment-specific Rewards

We add novel reward functions to the GridWorld, MemoryMaze and FetchPush environments. Table 1 shows an overview of all rewards used for evaluation. For the GridWorld and MemoryMaze environments, rewards are categorized as spatial (requiring positional and compositional knowledge) or observational (encodable from single observations). For the FetchPush environment, reward functions focus on spatial properties of the gripper and object as the observations contain solely global information.

## A.4. Model Architectures and Hyper-parameters

**Contrastive RL.** We base our implementation of CRL (Eysenbach et al., 2022) on the code provided by Zheng et al. (2024). For all experiments we use a two-layer multi-layer perceptron (MLP)-based policy with 256 hidden units each and ReLU activation. For image based environments, all images are scaled to $64 \times 64 \times 3$. We encode images to latent representation size of 256 using a convolutional neural network (CNN)-based encoder as proposed in Mnih et al. (2013). The encoders in the parametrized $Q$-value function consist of MLPs with two hidden layers with 256 units and ReLU activation, projecting down to a representation dimension of 32. Note that the encoder consists of two separate networks for state-action and goal encoding. For experiments using continuous actions, we use adaptive entropy regularization as proposed in Haarnoja et al. (2018) with a target entropy of 0.0. For discrete actions, we bound the maximum $D_{\text{KL}}$ between policy and a uniform prior policy by 1 via minimization ("free-bits" trick proposed in Kingma et al. (2016) and used by Hafner et al. (2025)). Batch-sizes vary depending on the problem and are mostly limited by memory: 128 for image-based tasks and 2048 for all other experiments. We try to maximize batch sizes as Zheng et al. (2024) showed that in general larger

---

[1] https://github.com/fbwurz/Hierarchical-Goal-Abstractions-via-Learned-Subset-Relations

| Environment | Category | Rewards | Description |
|---|---|---|---|
| **GridWorld** | Spatial | Individual rooms
Disjoint rooms (two rooms)
Regions (top, down, left, right) | Requires spatial knowledge |
| | Observational | Room centers
Shaped rooms (elliptical, polygonal, . . . )
Close to wall
Passages (between rooms) | Possible to encode by a single observation |
| **MemoryMaze** | Spatial | Individual rooms
Disjoint rooms (two rooms) | Requires spatial knowledge |
| | Observational | Looking at walls (any wall, red wall)
Looking at colored corners (blue, yellow) | Possible to encode by a single observation |
| **FetchPush** | Gripper-based | Off table
On table border
In table region (top, left, bottom, right) | Focus on gripper representation encoding |
| | Object-based | Off table
On table border
In table region (top, left, bottom, right) | Focus on object representation encoding |

*Table 1.* Environment-specific reward functions.

batch sizes are desired for CRL.

**$\beta$-VAE.** As a baseline, we train $\beta$-VAEs (Higgins et al., 2017; Nair et al., 2018) to encode observations into latent goals, using a latent dimension of 4 and a $\beta$ of 10. Following Nair et al. (2018), we use Euclidean distance in latent space as similarity measure. We obtain a similarity function $\text{Sim}(\mathbf{g}, \mathbf{g}') = \exp(-\|\mathbf{g} - \mathbf{g}'\|)$ resulting in high similarity for low distance and low similarity for high distances. This way, the similarity is bounded and suitable for encoding reward functions using Equation 3.

**DGR.** Dual goal representations (DGR) are based on the idea of using a relative perspective, representing a goal as the distance to all possible states (Park et al., 2026). This enables a set-based view on the goal, where the set of all states is related to the goal, where similarity is based on temporal proximity. Similarity is defined as $\text{Sim}(\mathbf{s}, \mathbf{s}') = \psi(\mathbf{s})^T \phi(\mathbf{s}')$ and the representation is learned with the contrastive learning objective used in CRL (Eysenbach et al., 2022). We use $\phi(\mathbf{s})$ as goal representation. Due to the unbounded latent space, we employ an $L2$ regularization during optimization of abstract goals.

**TRA.** Temporal representation alignment (TRA) (Myers et al., 2026) uses the same bilinear parametrization of the similarity function as used in DGR but uses an InfoNCE loss to learn the representations. Again, we use the same $L2$ regularization used for DGR when optimizing abstract goals.

**VIP.** Value implicit pre-training (VIP) (Ma et al., 2023) uses a metric based embedding $V(\mathbf{s}, \mathbf{s}') = \|\phi(\mathbf{s}) - \phi(\mathbf{s}')\|_2$ learnt from temporal distance. To transform the distance measure into a compatibility measure we use $\text{Sim}(\mathbf{s}, \mathbf{s}') = e^{-V(\mathbf{s}, \mathbf{s}')}$ and use an $L2$ regularization to deal with the unbounded latent space. Again, we use $\phi(\mathbf{s})$ as goal representation.

**Goal abstraction.** We encode sequences with a gated recurrent unit (Cho et al., 2014) with a hidden state size of 256. After encoding the whole sequence we use the last hidden state and use a discrete VAE (Hafner et al., 2022; 2025) using two hidden layers to project the RNN hidden state to a multi-categorical distribution. We use 16 categories with 16 possible values, resulting in a one-hot encoding of shape $16 \times 16$. For image-based observations we first encode the images using the CNN described in Mnih et al. (2013) to obtain a 256 dimensional encoding and apply our architecture to the encoded images. The energy function is parametrized as a simple 3 layer neural network with ReLU activation and a hidden dimension of 256. We use binary cross entropy as loss to train the energy function and a fixed batch size of 256 for all experiments.

**Reward encoding.** While rewards are translated into goals $\mathbf{g}_\vartheta$ by optimizing (3), we drop the negative feedback term when encoding observational goals in the FetchPush environment, because contrastive learning would eliminate static background

information in this case and thereby destroy almost all observational information.

To practically encode rewards into goals we parametrize the goal by a simple neural network, receiving a zero-vector of shape $64$ as input and outputting a goal in the corresponding goal space. For image-based goals, we use three transposed convolution layers, reversing the architecture of the CNN encoder. For the LiDAR observations and our approach we use a simple three layer MLP. Discretization for our goals is achieved by a final categorical distribution.

**Network sizes.** We ensured that network sizes are comparable between different goal representations by increasing/decreasing the width of hidden layers accordingly to match the number of trainable parameters.

### A.5. Training procedure and evaluation

We split training into a pre-training phase, where the policy and goal representation are pre-trained, and a fine-tuning phase, where we optimize over goals to encode reward functions. In the pre-training phase, we collect interactions in the environments for $16$ steps while performing one training step as a trade-off between sample efficiency and speed. The agents are pre-trained on the original, single observation goal reaching task where observations are provided by the environment. For GridWorld and FetchPush, pre-training lasts for $1000000$ environment steps. For MemoryMaze, we use $500000$ environment steps. During the first $50000$ environment steps, we use additional losses to stabilize training and improve performance. First, we use a reconstruction objective, reconstructing single observations from the encoded goals. This helps with early learning, especially in image-based environments where otherwise the subset relation takes a long time to find good representations. Note that we fully transition to contrastive learning later in training, as reconstruction hinders convergence at some point. Furthermore, we regularize the discrete latent space by imposing a $D_{\text{KL}}$ constraint on the categorical distribution, effectively regularizing it towards a uniform distribution. This constraint loosens over the course of the first $50000$ environment steps. This is required as during early training the encoder may collapse.

We evaluate what rewards each considered goal representation is able to represent as well as how well the pre-trained and frozen policies perform. For each novel reward function, the agent interacts with the environment for a total of $20000$ steps. The environment steps are split up into episodes of $50$ steps. The resulting goal representations are evaluated based on heatmaps as well as by the alignment criterion. The agents' behavior is assessed by the performance criteria given Figure 4 and depicted in Figure 7. After learning the reward function, the agent is tasked to reach the optimized goal $100$ times starting from different initial states and with a budget of $50$ environment steps. For each reward function, three runs are performed and averaged. Then reward groups are averaged to obtain the final plot. The star plot in Figure 7 min-max normalizes the performance metrics such that $0$ is the worst observed performance and $1$ is the best observed performance over all rewards in an environment. Attainment-dependent metrics are scaled by the attainment value such that non-attaining episodes result in worst performance ($0$).

## B. Additional Results

Here we present additional results on the representative power of our goal representation as well as down-stream policy performance.

### B.1. Numerical Results

Comprehensive numerical results across all baselines and environments are detailed in Table 2, Table 3, and Table 4. Over all experiments, our approach works most consistently in terms of goal attainment and reward alignment across all three environments. TRA mirrors the performance of DGR on most metrics. This is expected given its shared bilinear structure. While both show highly competitive reward alignment scores, they fall short on goal-conditioned policy performance, particularly in the FetchPush environment. This discrepancy further supports our hypothesis that while bilinear models capture the abstract reward structure, they lack an effective mechanism for hierarchy traversal during policy training. The $\beta$-VAE performs reasonably in GridWorld but degrades in more complex environments. In MemoryMaze and FetchPush it consistently falls below our approach on both goal attainment and reward alignment. We observe a similar trend with the VIP baseline: while it yields only moderate results on GridWorld and MemoryMaze and generally suffers from low goal attainment scores, its performance is surprisingly good in the FetchPush environment.

**Observation Group**

| Model | Goal Att. | Total Rew. | Final Rew. | Final Streak | Max. Streak | Avg. Streak | Goal Speed | Rew. Align. |
|---|---|---|---|---|---|---|---|---|
| Abstract (relabel only) | $0.55 \pm 0.20$ | $0.35 \pm 0.17$ | $0.24 \pm 0.13$ | $0.14 \pm 0.10$ | $0.25 \pm 0.14$ | $0.22 \pm 0.14$ | $0.31 \pm 0.15$ | $0.52 \pm 0.23$ |
| Abstract (Eq. 3) | $\mathbf{0.66 \pm 0.16}$ | $\mathbf{0.39 \pm 0.16}$ | $0.28 \pm 0.13$ | $\mathbf{0.16 \pm 0.10}$ | $\mathbf{0.27 \pm 0.15}$ | $\mathbf{0.24 \pm 0.15}$ | $\mathbf{0.35 \pm 0.14}$ | $0.52 \pm 0.24$ |
| Abstract (Eq. 1) | $0.56 \pm 0.19$ | $0.28 \pm 0.18$ | $0.17 \pm 0.14$ | $0.09 \pm 0.10$ | $0.20 \pm 0.16$ | $0.18 \pm 0.14$ | $0.31 \pm 0.07$ | $0.50 \pm 0.19$ |
| DGR | $0.53 \pm 0.23$ | $0.27 \pm 0.13$ | $0.15 \pm 0.09$ | $0.07 \pm 0.06$ | $0.19 \pm 0.11$ | $0.15 \pm 0.11$ | $0.31 \pm 0.15$ | $0.71 \pm 0.30$ |
| TRA | $0.49 \pm 0.22$ | $0.32 \pm 0.22$ | $0.23 \pm 0.18$ | $0.16 \pm 0.14$ | $0.26 \pm 0.18$ | $0.22 \pm 0.18$ | $0.32 \pm 0.16$ | $\mathbf{0.76 \pm 0.33}$ |
| VIP | $0.46 \pm 0.26$ | $0.29 \pm 0.20$ | $0.21 \pm 0.26$ | $0.11 \pm 0.09$ | $0.23 \pm 0.13$ | $0.20 \pm 0.11$ | $0.31 \pm 0.16$ | $0.37 \pm 0.12$ |
| $\beta$-VAE | $0.56 \pm 0.21$ | $0.35 \pm 0.19$ | $\mathbf{0.29 \pm 0.19}$ | $0.15 \pm 0.12$ | $0.25 \pm 0.16$ | $0.22 \pm 0.16$ | $0.33 \pm 0.18$ | $0.34 \pm 0.15$ |
| Observation | $0.57 \pm 0.19$ | $0.25 \pm 0.15$ | $0.17 \pm 0.15$ | $0.08 \pm 0.07$ | $0.18 \pm 0.11$ | $0.16 \pm 0.11$ | $0.31 \pm 0.11$ | $0.33 \pm 0.16$ |
| Random | $0.36 \pm 0.21$ | $0.17 \pm 0.08$ | $0.07 \pm 0.04$ | $0.05 \pm 0.04$ | $0.16 \pm 0.11$ | $0.14 \pm 0.11$ | $0.25 \pm 0.11$ | $0.31 \pm 0.15$ |

**Spatial Group**

| Model | Goal Att. | Total Rew. | Final Rew. | Final Streak | Max. Streak | Avg. Streak | Goal Speed | Rew. Align. |
|---|---|---|---|---|---|---|---|---|
| Abstract (relabel only) | $0.50 \pm 0.06$ | $0.45 \pm 0.04$ | $0.28 \pm 0.05$ | $0.21 \pm 0.05$ | $0.37 \pm 0.07$ | $0.34 \pm 0.06$ | $0.31 \pm 0.13$ | $0.86 \pm 0.06$ |
| Abstract (Eq. 3) | $\mathbf{0.63 \pm 0.10}$ | $\mathbf{0.62 \pm 0.09}$ | $\mathbf{0.48 \pm 0.14}$ | $\mathbf{0.41 \pm 0.15}$ | $\mathbf{0.51 \pm 0.13}$ | $\mathbf{0.50 \pm 0.13}$ | $0.33 \pm 0.14$ | $0.88 \pm 0.05$ |
| Abstract (Eq. 1) | $0.56 \pm 0.08$ | $0.54 \pm 0.08$ | $0.38 \pm 0.10$ | $0.31 \pm 0.12$ | $0.44 \pm 0.11$ | $0.42 \pm 0.12$ | $0.29 \pm 0.12$ | $0.87 \pm 0.05$ |
| DGR | $0.58 \pm 0.16$ | $0.57 \pm 0.16$ | $0.43 \pm 0.23$ | $0.39 \pm 0.24$ | $0.49 \pm 0.18$ | $0.48 \pm 0.19$ | $0.34 \pm 0.16$ | $0.98 \pm 0.02$ |
| TRA | $0.57 \pm 0.20$ | $0.56 \pm 0.21$ | $0.44 \pm 0.29$ | $0.41 \pm 0.30$ | $0.50 \pm 0.23$ | $0.49 \pm 0.25$ | $\mathbf{0.36 \pm 0.18}$ | $\mathbf{0.99 \pm 0.02}$ |
| VIP | $0.51 \pm 0.22$ | $0.46 \pm 0.23$ | $0.36 \pm 0.26$ | $0.31 \pm 0.22$ | $0.40 \pm 0.20$ | $0.39 \pm 0.20$ | $0.32 \pm 0.12$ | $0.47 \pm 0.12$ |
| $\beta$-VAE | $0.51 \pm 0.10$ | $0.45 \pm 0.09$ | $0.27 \pm 0.10$ | $0.22 \pm 0.11$ | $0.38 \pm 0.11$ | $0.35 \pm 0.12$ | $0.31 \pm 0.14$ | $0.47 \pm 0.11$ |
| Observation | $0.37 \pm 0.15$ | $0.23 \pm 0.20$ | $0.12 \pm 0.16$ | $0.11 \pm 0.15$ | $0.22 \pm 0.18$ | $0.21 \pm 0.18$ | $0.31 \pm 0.12$ | $0.43 \pm 0.13$ |
| Random | $0.27 \pm 0.13$ | $0.24 \pm 0.12$ | $0.10 \pm 0.08$ | $0.11 \pm 0.09$ | $0.27 \pm 0.14$ | $0.26 \pm 0.14$ | $0.28 \pm 0.13$ | $0.42 \pm 0.13$ |

*Table 2.* Comparison of all persuadability metrics on the GridWorld environment.

## B.2. Sequential vs. Non-sequential Data

To investigate the importance of sequential order, we trained the sequence encoder on both the original sequential data and a shuffled, non-sequential version. Figure 8 demonstrates that both learned representations yield similar performance across all metrics in the downstream persuadability task. This indicates that the RNN does not require the temporal order of the observations to encode goals. Although alternative time-invariant set encoders like DeepSets (Zaheer et al., 2017) are applicable, we employ an RNN-based encoder for its simplicity and efficiency.

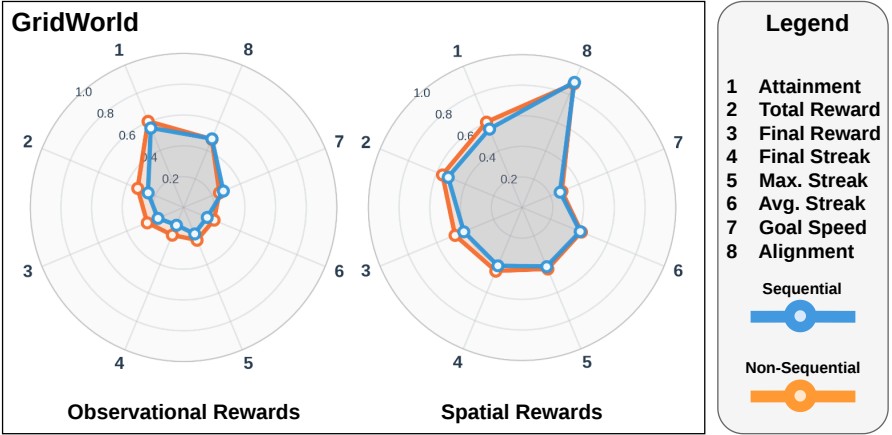

*Figure 8.* Agent performance on novel rewards for the GridWorld environment using sequential and non-sequential (permuted in time) data to learn the representation.

## B.3. Additional Goal Heatmaps

Next we consider the MemoryMaze environment where we learn abstract goals from image observations by using interaction with the environment. We split up the visualization into approaches using asymmetric temporal information (Figure 9) and non-temporal/symmetric temporal information (Figure 10). For the former, we compare our approach (using the induced

**Observation Group**

| Model | Goal Att. | Total Rew. | Final Rew. | Final Streak | Max. Streak | Avg. Streak | Goal Speed | Rew. Align. |
|---|---|---|---|---|---|---|---|---|
| Abstract (relabel only) | $\mathbf{0.58 \pm 0.22}$ | $0.36 \pm 0.25$ | $\mathbf{0.30 \pm 0.30}$ | $0.22 \pm 0.25$ | $0.24 \pm 0.22$ | $0.20 \pm 0.18$ | $0.29 \pm 0.28$ | $0.33 \pm 0.08$ |
| Abstract (Eq. 3) | $0.57 \pm 0.21$ | $0.35 \pm 0.24$ | $0.27 \pm 0.30$ | $0.19 \pm 0.23$ | $0.23 \pm 0.22$ | $0.19 \pm 0.16$ | $0.28 \pm 0.30$ | $0.35 \pm 0.16$ |
| Abstract (Eq. 1) | $0.53 \pm 0.25$ | $0.33 \pm 0.26$ | $0.25 \pm 0.27$ | $0.22 \pm 0.26$ | $0.23 \pm 0.23$ | $0.21 \pm 0.20$ | $0.25 \pm 0.27$ | $0.37 \pm 0.10$ |
| DGR | $0.54 \pm 0.25$ | $\mathbf{0.38 \pm 0.26}$ | $0.30 \pm 0.32$ | $\mathbf{0.26 \pm 0.32}$ | $\mathbf{0.27 \pm 0.24}$ | $\mathbf{0.23 \pm 0.21}$ | $0.27 \pm 0.30$ | $0.45 \pm 0.11$ |
| TRA | $0.51 \pm 0.24$ | $0.33 \pm 0.26$ | $0.26 \pm 0.31$ | $0.22 \pm 0.31$ | $0.23 \pm 0.25$ | $0.20 \pm 0.21$ | $0.27 \pm 0.28$ | $\mathbf{0.46 \pm 0.09}$ |
| VIP | $0.50 \pm 0.26$ | $0.31 \pm 0.26$ | $0.24 \pm 0.27$ | $0.20 \pm 0.27$ | $0.22 \pm 0.23$ | $0.18 \pm 0.19$ | $0.28 \pm 0.28$ | $0.26 \pm 0.10$ |
| $\beta$-VAE | $0.51 \pm 0.26$ | $0.34 \pm 0.27$ | $0.29 \pm 0.32$ | $0.25 \pm 0.29$ | $0.25 \pm 0.24$ | $0.22 \pm 0.20$ | $0.27 \pm 0.28$ | $0.34 \pm 0.05$ |
| Observation | $0.46 \pm 0.28$ | $0.26 \pm 0.28$ | $0.23 \pm 0.33$ | $0.17 \pm 0.26$ | $0.20 \pm 0.23$ | $0.16 \pm 0.18$ | $\mathbf{0.31 \pm 0.29}$ | $0.33 \pm 0.06$ |
| Random | $0.32 \pm 0.31$ | $0.21 \pm 0.27$ | $0.16 \pm 0.28$ | $0.17 \pm 0.30$ | $0.19 \pm 0.25$ | $0.16 \pm 0.22$ | $0.26 \pm 0.28$ | $0.31 \pm 0.01$ |

**Spatial Group**

| Model | Goal Att. | Total Rew. | Final Rew. | Final Streak | Max. Streak | Avg. Streak | Goal Speed | Rew. Align. |
|---|---|---|---|---|---|---|---|---|
| Abstract (relabel only) | $\mathbf{0.28 \pm 0.15}$ | $\mathbf{0.25 \pm 0.13}$ | $0.11 \pm 0.08$ | $\mathbf{0.14 \pm 0.10}$ | $\mathbf{0.23 \pm 0.12}$ | $\mathbf{0.23 \pm 0.12}$ | $0.22 \pm 0.11$ | $0.65 \pm 0.08$ |
| Abstract (Eq. 3) | $0.26 \pm 0.14$ | $0.22 \pm 0.13$ | $0.10 \pm 0.09$ | $0.12 \pm 0.10$ | $0.22 \pm 0.12$ | $0.21 \pm 0.11$ | $0.22 \pm 0.12$ | $0.60 \pm 0.11$ |
| Abstract (Eq. 1) | $0.25 \pm 0.14$ | $0.21 \pm 0.13$ | $0.09 \pm 0.08$ | $0.12 \pm 0.11$ | $0.22 \pm 0.13$ | $0.21 \pm 0.13$ | $0.21 \pm 0.12$ | $0.74 \pm 0.10$ |
| DGR | $0.26 \pm 0.16$ | $0.23 \pm 0.15$ | $0.11 \pm 0.11$ | $0.13 \pm 0.12$ | $0.22 \pm 0.13$ | $0.21 \pm 0.12$ | $\mathbf{0.22 \pm 0.10}$ | $\mathbf{0.88 \pm 0.03}$ |
| TRA | $0.26 \pm 0.16$ | $0.23 \pm 0.16$ | $\mathbf{0.12 \pm 0.12}$ | $0.14 \pm 0.16$ | $0.23 \pm 0.15$ | $0.21 \pm 0.16$ | $0.22 \pm 0.11$ | $\mathbf{0.88 \pm 0.03}$ |
| VIP | $0.24 \pm 0.11$ | $0.19 \pm 0.08$ | $0.07 \pm 0.04$ | $0.08 \pm 0.05$ | $0.18 \pm 0.08$ | $0.17 \pm 0.08$ | $0.22 \pm 0.12$ | $0.26 \pm 0.10$ |
| $\beta$-VAE | $0.27 \pm 0.15$ | $0.22 \pm 0.15$ | $0.11 \pm 0.10$ | $0.13 \pm 0.12$ | $0.22 \pm 0.14$ | $0.21 \pm 0.14$ | $0.22 \pm 0.13$ | $0.53 \pm 0.08$ |
| Observation | $0.21 \pm 0.13$ | $0.16 \pm 0.13$ | $0.06 \pm 0.08$ | $0.08 \pm 0.11$ | $0.17 \pm 0.14$ | $0.16 \pm 0.14$ | $0.22 \pm 0.13$ | $0.46 \pm 0.15$ |
| Random | $0.19 \pm 0.12$ | $0.16 \pm 0.11$ | $0.06 \pm 0.06$ | $0.07 \pm 0.08$ | $0.17 \pm 0.11$ | $0.15 \pm 0.10$ | $0.22 \pm 0.11$ | $0.51 \pm 0.10$ |

*Table 3.* Comparison of all persuadability metrics on the MemoryMaze environment.

partial order) as well as DGR and TRA, which both use a bilinear similarity function. The latter includes observational goals, the $\beta$-VAE baseline to represent standard latent bottleneck approaches, and VIP (using a symmetric similarity function).

The asymmetric approaches are able to encode all considered goals, ranging from observational goals like colored corners or walls to more abstract spatial goals combining different rooms. For the non-temporal and symmetric methods, the $\beta$-VAE is able to capture many aspects of the observation goals and most abstract goals; however, it shows broad, noisy activity levels throughout space without clear delimitations due to its symmetric, distance-based similarity measure. Observational goals struggle with encoding spatial concepts like a specific room but are able to encode some observational properties like colored walls or corners. VIP struggles to capture the relevant rewards.

For the FetchPush environment, only spatial goals regarding the gripper and object are considered. We again split the visualization into asymmetric (Figure 11) and non-temporal/symmetric approaches (Figure 12). While our method is able to encode both gripper-based (specific regions, off-table) and object-based (specific regions, off-table) rewards, the other approaches fail to capture the full spectrum of reward functions. Consistent with our main text analysis, DGR and TRA can encode object-specific rewards with high alignment but struggle to capture gripper-related rewards. Observational goals are not able to capture any meaningful structure while VIP solves all tasks comparable to our approach. Finally, the $\beta$-VAE is able to represent some gripper-based rewards but struggles entirely with the object-based rewards.

### B.4. Abstract Goal Relabeling

For our goal relabelling during contrastive learning we use entire sequences of observations to be relabeled as a single, more abstract goal. In $10\%$ of the cases we replace the relabeled sequence goal by a more abstract goal by further optimizing our energy function with our energy-driven and reward-driven goal traversal approach proposed in Section 2. This extra optimization enables the policy to be trained with slightly more abstract goals than the ones that were actually experienced. Figure 13 shows the difference in performance of the policies with and without this extra optimization step. Especially, for spatial and object-based reward functions the extra abstraction also leads to an extra improvement in performance.

### B.5. Transitivity Analysis (GridWorld)

To assess the quality of the abstractions contained in the energy function, we check for internal consistency, in particular transitivity, i.e. if $g_0 \preceq g_1$ and $g_1 \preceq g_2$ then it should follow that $g_0 \preceq g_2$. Starting from a goal $g_0$ grounded in the agent's

**Gripper Group**

| Model | Goal Att. | Total Rew. | Final Rew. | Final Streak | Max. Streak | Avg. Streak | Goal Speed | Rew. Align. |
|---|---|---|---|---|---|---|---|---|
| Abstract (relabel only) | $0.59 \pm 0.35$ | $0.33 \pm 0.22$ | $0.24 \pm 0.23$ | $0.11 \pm 0.12$ | $0.20 \pm 0.16$ | $0.15 \pm 0.13$ | $0.38 \pm 0.34$ | $0.48 \pm 0.12$ |
| Abstract (Eq. 3) | $0.70 \pm 0.34$ | $\mathbf{0.53 \pm 0.29}$ | $\mathbf{0.45 \pm 0.30}$ | $\mathbf{0.29 \pm 0.23}$ | $\mathbf{0.39 \pm 0.27}$ | $\mathbf{0.33 \pm 0.24}$ | $0.48 \pm 0.33$ | $0.37 \pm 0.12$ |
| Abstract (Eq. 1) | $0.61 \pm 0.32$ | $0.34 \pm 0.21$ | $0.23 \pm 0.21$ | $0.09 \pm 0.09$ | $0.20 \pm 0.16$ | $0.15 \pm 0.11$ | $0.41 \pm 0.31$ | $0.42 \pm 0.14$ |
| DGR | $0.56 \pm 0.34$ | $0.37 \pm 0.22$ | $0.25 \pm 0.20$ | $0.11 \pm 0.11$ | $0.21 \pm 0.15$ | $0.18 \pm 0.13$ | $0.34 \pm 0.33$ | $0.49 \pm 0.13$ |
| TRA | $0.62 \pm 0.35$ | $0.41 \pm 0.25$ | $0.25 \pm 0.20$ | $0.12 \pm 0.12$ | $0.26 \pm 0.21$ | $0.21 \pm 0.17$ | $0.39 \pm 0.33$ | $0.57 \pm 0.12$ |
| VIP | $\mathbf{0.71 \pm 0.36}$ | $0.51 \pm 0.33$ | $0.42 \pm 0.35$ | $0.25 \pm 0.24$ | $0.36 \pm 0.29$ | $0.30 \pm 0.25$ | $\mathbf{0.50 \pm 0.34}$ | $\mathbf{0.62 \pm 0.11}$ |
| $\beta$-VAE | $0.55 \pm 0.33$ | $0.33 \pm 0.22$ | $0.19 \pm 0.22$ | $0.09 \pm 0.14$ | $0.19 \pm 0.21$ | $0.15 \pm 0.16$ | $0.34 \pm 0.34$ | $0.53 \pm 0.20$ |
| Observation | $0.50 \pm 0.28$ | $0.25 \pm 0.16$ | $0.12 \pm 0.13$ | $0.04 \pm 0.05$ | $0.14 \pm 0.16$ | $0.10 \pm 0.11$ | $0.34 \pm 0.34$ | $0.54 \pm 0.19$ |
| Random | $0.47 \pm 0.34$ | $0.25 \pm 0.22$ | $0.15 \pm 0.20$ | $0.06 \pm 0.12$ | $0.16 \pm 0.22$ | $0.11 \pm 0.16$ | $0.35 \pm 0.36$ | $0.47 \pm 0.17$ |

**Object Group**

| Model | Goal Att. | Total Rew. | Final Rew. | Final Streak | Max. Streak | Avg. Streak | Goal Speed | Rew. Align. |
|---|---|---|---|---|---|---|---|---|
| Abstract (relabel only) | $0.55 \pm 0.29$ | $0.37 \pm 0.20$ | $0.25 \pm 0.17$ | $0.17 \pm 0.12$ | $0.26 \pm 0.15$ | $0.25 \pm 0.16$ | $0.36 \pm 0.30$ | $0.56 \pm 0.16$ |
| Abstract (Eq. 3) | $0.71 \pm 0.32$ | $\mathbf{0.55 \pm 0.30}$ | $0.53 \pm 0.32$ | $\mathbf{0.45 \pm 0.29}$ | $\mathbf{0.48 \pm 0.27}$ | $\mathbf{0.46 \pm 0.28}$ | $\mathbf{0.53 \pm 0.29}$ | $0.55 \pm 0.10$ |
| Abstract (Eq. 1) | $0.59 \pm 0.27$ | $0.50 \pm 0.23$ | $0.36 \pm 0.27$ | $0.32 \pm 0.31$ | $0.40 \pm 0.29$ | $0.41 \pm 0.29$ | $0.39 \pm 0.30$ | $0.67 \pm 0.15$ |
| DGR | $0.56 \pm 0.33$ | $0.40 \pm 0.26$ | $0.33 \pm 0.25$ | $0.25 \pm 0.20$ | $0.32 \pm 0.21$ | $0.30 \pm 0.22$ | $0.40 \pm 0.32$ | $0.68 \pm 0.14$ |
| TRA | $0.56 \pm 0.35$ | $0.39 \pm 0.29$ | $0.31 \pm 0.27$ | $0.26 \pm 0.23$ | $0.33 \pm 0.25$ | $0.32 \pm 0.26$ | $0.42 \pm 0.32$ | $\mathbf{0.76 \pm 0.21}$ |
| VIP | $\mathbf{0.71 \pm 0.28}$ | $0.50 \pm 0.26$ | $0.44 \pm 0.29$ | $0.37 \pm 0.25$ | $0.43 \pm 0.23$ | $0.42 \pm 0.24$ | $0.52 \pm 0.28$ | $0.32 \pm 0.17$ |
| $\beta$-VAE | $0.44 \pm 0.36$ | $0.35 \pm 0.29$ | $0.27 \pm 0.33$ | $0.23 \pm 0.31$ | $0.28 \pm 0.31$ | $0.27 \pm 0.29$ | $0.30 \pm 0.35$ | $0.46 \pm 0.16$ |
| Observation | $0.28 \pm 0.32$ | $0.18 \pm 0.15$ | $0.10 \pm 0.18$ | $0.09 \pm 0.17$ | $0.14 \pm 0.19$ | $0.14 \pm 0.18$ | $0.21 \pm 0.35$ | $0.56 \pm 0.17$ |
| Random | $0.17 \pm 0.37$ | $0.09 \pm 0.19$ | $0.08 \pm 0.19$ | $0.10 \pm 0.23$ | $0.11 \pm 0.23$ | $0.11 \pm 0.24$ | $0.17 \pm 0.37$ | $0.52 \pm 0.19$ |

*Table 4.* Comparison of all persuadability metrics on the FetchPush-Image environment.

experience, we generate a sequence $(g_0, g_1, \ldots, g_t)$ of goals iteratively. We evaluate this procedure over 50 independent runs with 6 abstraction steps per run. Subsequent goals are obtained through optimization. Consequently, subsequent goals do not necessarily correspond to a particular experienced sequence but, for example, might represent a disjoint set of observations that share some common feature (e.g, being close to a wall, center of a room, corners, etc.). Since the energy function never experienced these more abstract goals during training, we naturally expect a degradation in goal quality as more steps are taken to obtain more and more abstract goals. We analyze goals generated by the energy-driven as well as the reward-driven traversal approach described in Section 2. In practice, we generate 50 candidate goals for each abstraction step; for the reward-guided method for each candidate a pool of 64 random concrete goals followed by a selection of the 8 highest rewarding goals is used for optimization. Both methods then optimize for more abstract goals using 5 gradient steps per iteration step.

Figure 14a indicates that reward-guided traversal yields a cleaner upper-triangular structure, consistent with a more stable transitive ordering, whereas the energy-driven variant departs earlier from this triangular pattern. The trend in Figure 14b shows that reward-guided traversal retains a larger fraction of previously positive states across steps, indicating stronger preservation of established support. A similar difference appears in Figure 14c: the reward-guided variant shows a more stable increase in spatial extent, while the energy-driven variant exhibits stronger drift in the growth of the active region. Moreover, Figure 14d shows that optimized energy decreases more rapidly for the energy-driven method, suggesting that abstraction becomes harder to optimize for as depth increases. Overall, both approaches indicate that the energy captures the partial order and the spatial extent of the optimized goals increase with more abstraction steps, matching our interpretation of abstraction. However, the outcome depends strongly on how the energy is optimized. The reward-guided traversal tends to preserve a cleaner structure, whereas the energy-driven traversal is more prone to drift as abstraction depth increases.

## C. Computational Overhead

The compared algorithms differ significantly in design, making a precise quantitative comparison of computational cost not straightforward. For a qualitative assessment, two components in our method introduce additional overhead relative to single-observation baselines. During representation training, we consider sets of states rather than single states, requiring subset sampling and contrastive negatives (see Alg. 1). During agent training, abstract hindsight relabeling requires optimization in goal space. Relabeling itself requires encoding a set of states into an abstract goal. The additional optimization steps differ in computation. Our energy-driven abstraction (Eq. 1) adds $S$ gradient steps to optimize the energy. The reward-guided

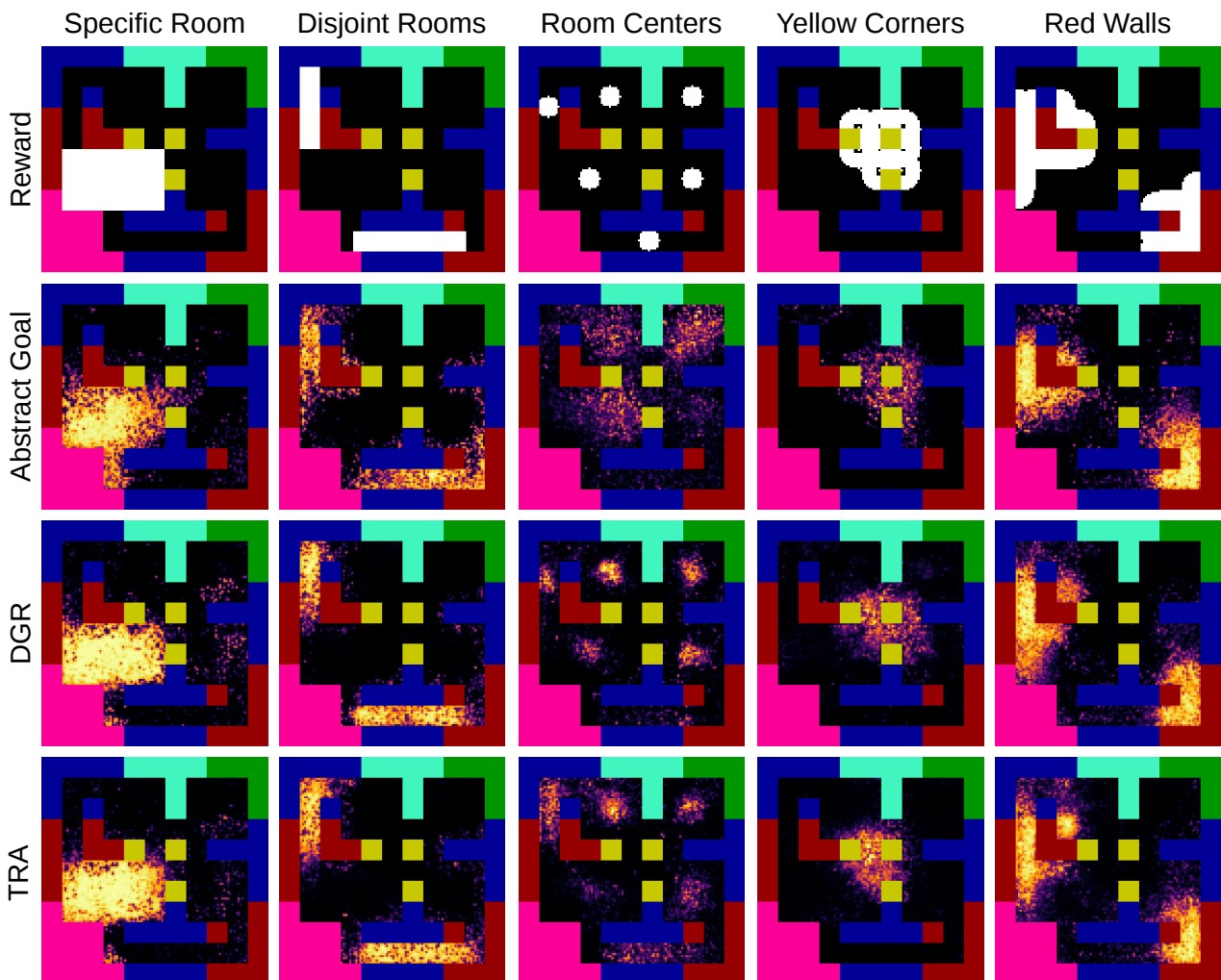

*Figure 9.* Encoded reward functions in the MemoryMaze for approaches using asymmetric temporal similarity. White regions in the top row indicate high rewarding states.

abstraction (using Eq. 3) uses one gradient-ascent step to score $C$ candidate goals, from which the top $K$ with highest compatibility are selected. Finally, $S$ gradient steps are performed to join those $K$ candidates into a more abstract goal.

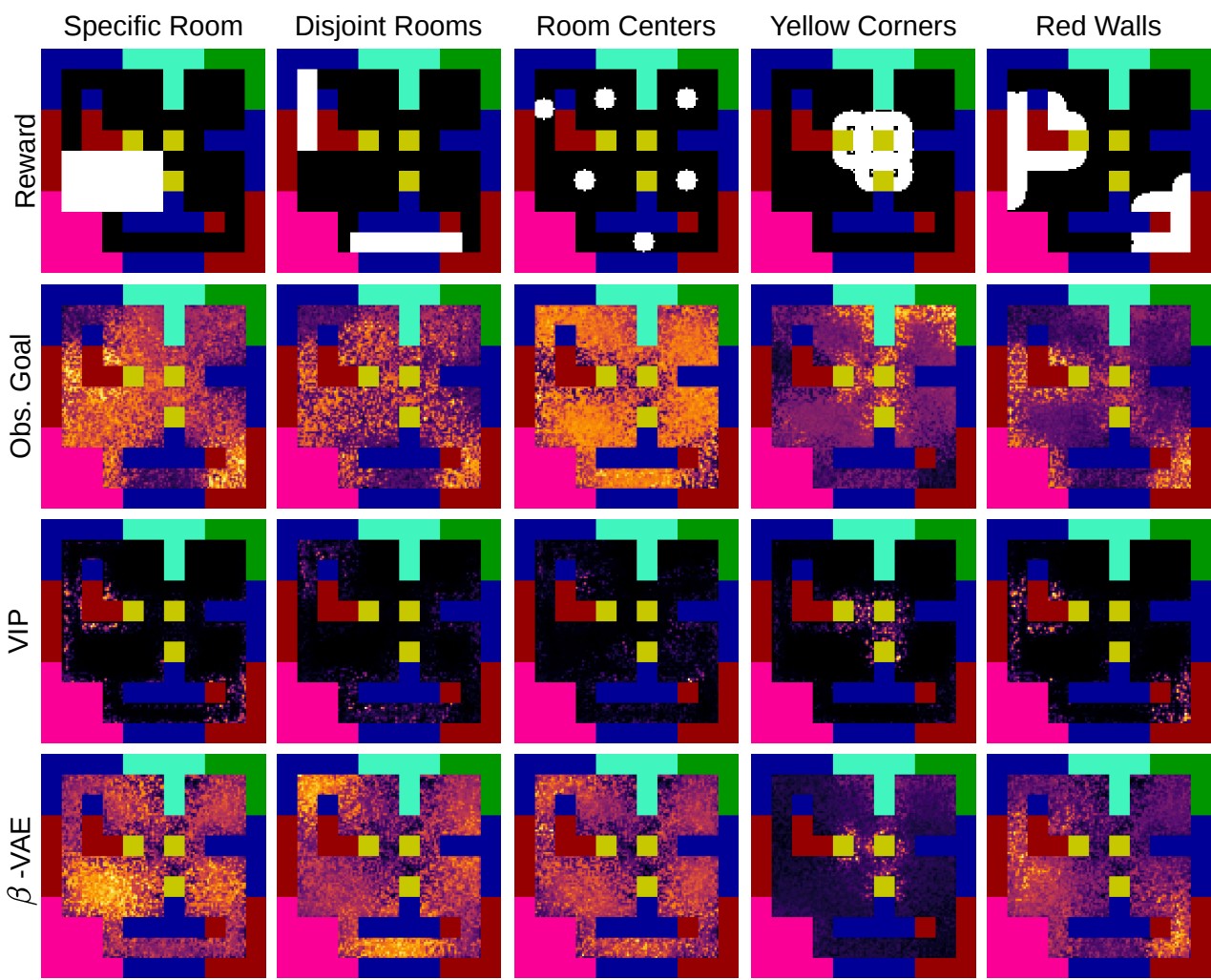

*Figure 10.* Encoded reward functions in the MemoryMaze environment for approaches using non-temporal information as well as symmetric temporal information. White regions in the top row indicate high rewarding states.

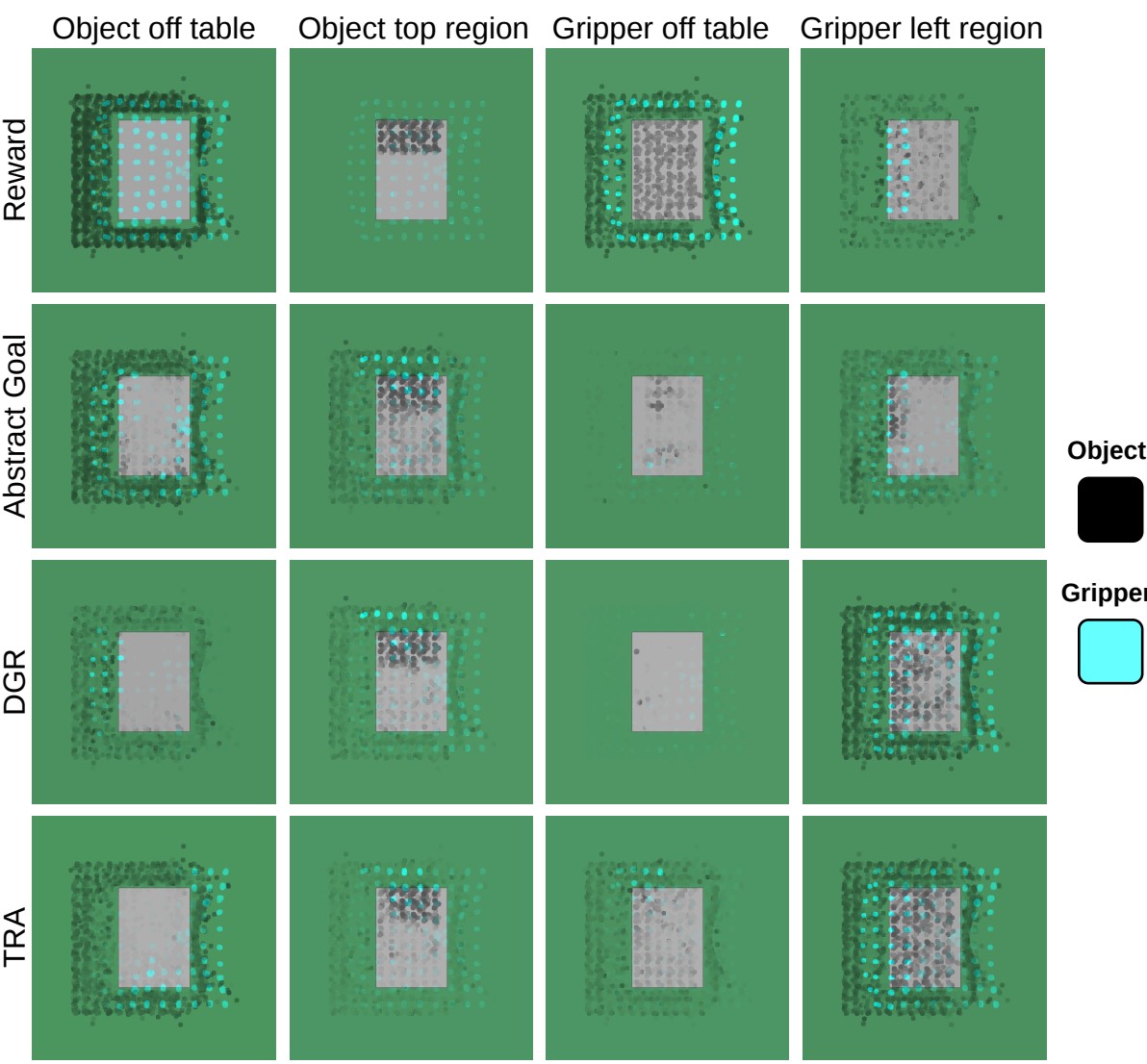

*Figure 11.* Encoded reward functions in the FetchPush environment (top-down view) for approaches using asymmetric temporal information. Black regions indicate high rewarding states for the object while cyan regions are used for gripper rewards. Our approach is able to represent both, gripper and object rewards.

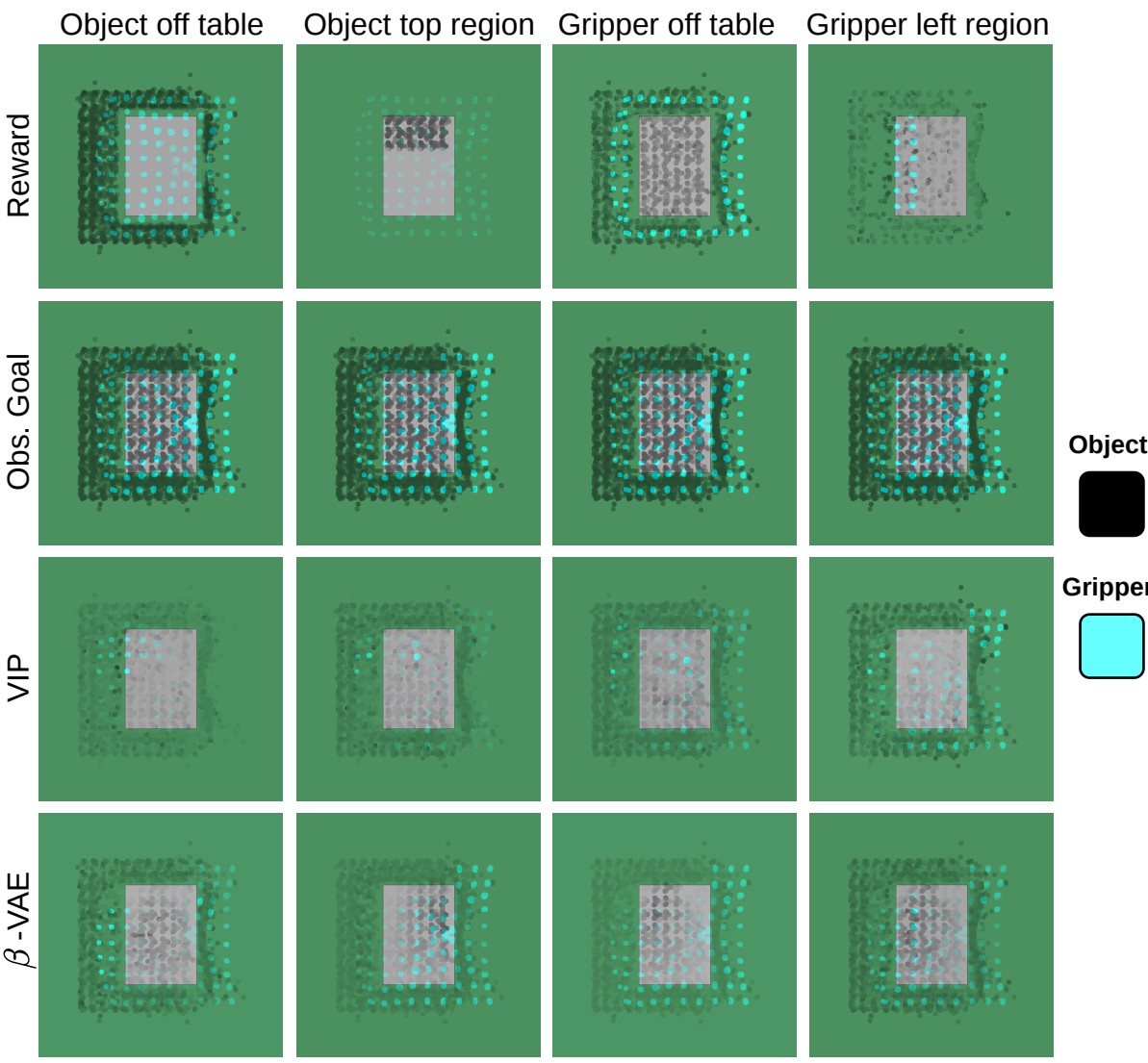

*Figure 12.* Encoded reward functions in the FetchPush environment (top-down view) for approaches using non-temporal and symmetric temporal information. Black regions indicate high rewarding states for the object while cyan regions are used for gripper rewards.

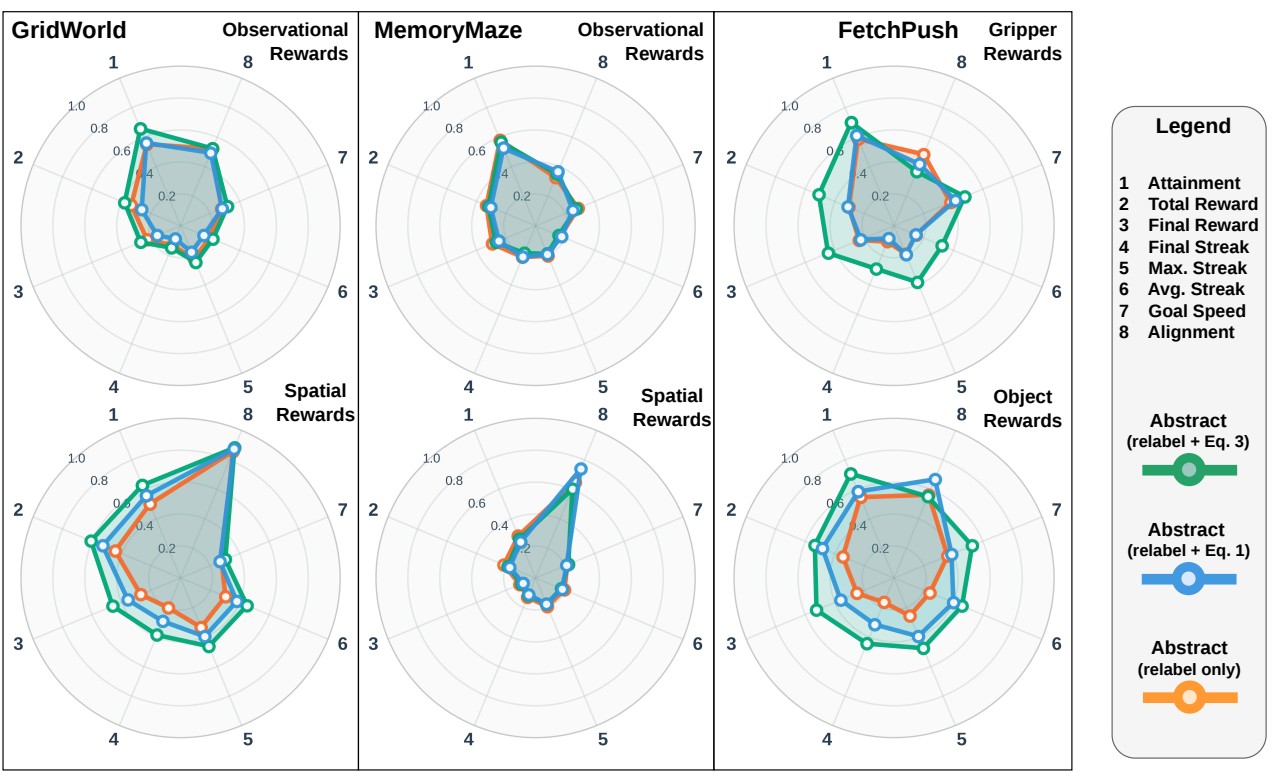

*Figure 13.* Agent performance on novel rewards for all considered environments using different relabeling strategies for our approach.

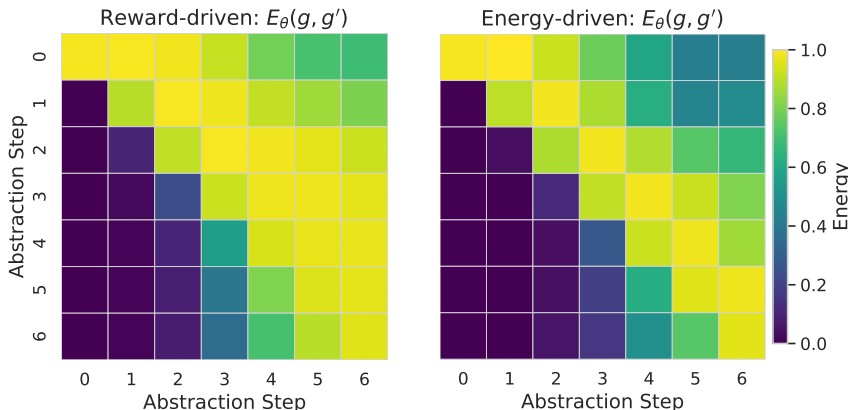

*(a)* Energy heatmaps for reward-driven and energy-driven hierarchy traversal. Higher values indicate stronger subset relations between goals.

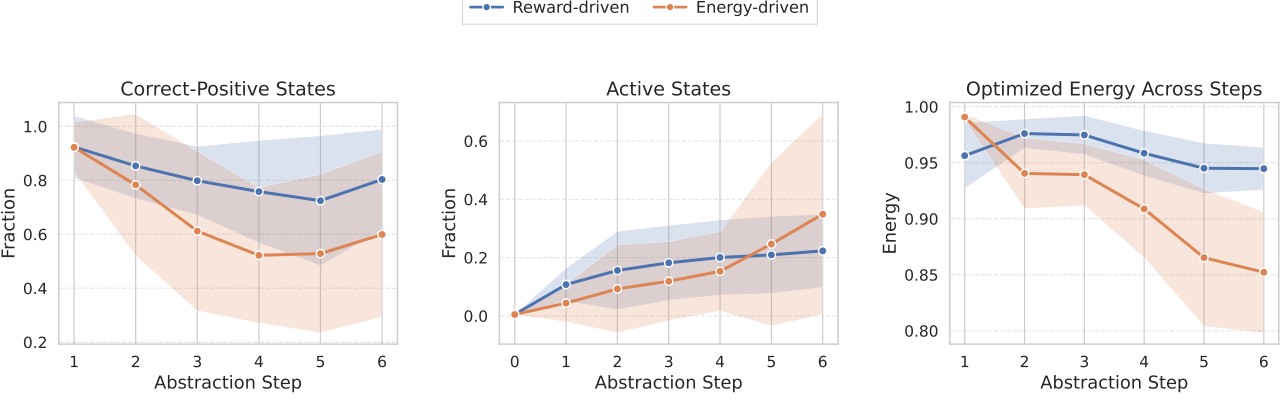

*(b)* Fraction of active states (spatial support of the goal) retained over abstraction steps. Values close to 1 indicate that the spatial support of the previous goal is preserved across abstraction steps.

*(c)* Spatial extent of the active region across abstraction steps. With more abstraction steps the goal region grows.

*(d)* Optimized energy of the selected abstract goals across abstraction steps. Subsequent goals are harder to optimize for.

*Figure 14.* Transitivity analysis comparing reward-driven vs. energy-driven hierarchy traversal. Top: pairwise energy matrices over the generated goal sequence, where a clean upper-triangular pattern indicates a more consistent hierarchy. Bottom: transitivity metrics across abstraction steps.

