# OpenReview forum: "Hierarchical Goal Abstractions via Learned Subset Relations"
_ICML.cc/2026/Conference — ICML 2026 regular_

### Official Review · Reviewer_EHQw · 2026-03-08

**Soundness:** 2
**Presentation:** 2
**Significance:** 3
**Originality:** 3
**Overall Recommendation:** 4
**Confidence:** 2

**Summary:**

This work focuses on the advancement of goal-conditioned RL. The authors propose constructing hierarchical goals that range from concrete to abstract by leveraging the subset inclusion relationships between trajectories. This framework allows agents to autonomously traverse the goal hierarchy, enabling generalization to novel reward functions and a deeper understanding of complex tasks.

**Compliance With Llm Reviewing Policy:**

Affirmed.

**Final Justification:**

The authors’ rebuttal has effectively addressed my main concerns, particularly regarding the strength and selection of baselines, which improves my confidence in the empirical evaluation and overall soundness of the work.

In terms of originality, the proposed idea appears potentially novel and interesting, and I will adjust my score accordingly in light of this.
While the significance appears promising, I still have some remaining concerns regarding the clarity of the implementation details and the potential computational overhead, as these aspects were not fully clarified in the rebuttal.
Overall, the rebuttal has positively influenced my evaluation and addressed my primary concerns.

**Please note that I am not an expert in this specific area, and my assessment may not fully reflect the originality of the work or its potential impact on the community. This should be taken into consideration when weighing my evaluation.**

**Key Questions For Authors:**

See weaknesses

**Limitations:**

yes

**Strengths And Weaknesses:**

While I am not an expert in this field, here is what I think:

**Strengths:**
- The idea of learning subset relationships between goals to help the agent grasp different levels of abstraction is both compelling and intuitive.

- The experimental results show a substantial improvement compared to the baselines.

---

**Weaknesses:**
- The primary baselines used, specifically Obs. and $\beta$-VAE, date back to 2017 or 2018. There is a concern that these baselines may be too weak or outdated to represent the current SOTA. Furthermore, the experimental evaluation is not sufficiently comprehensive or in-depth.

- The additional computational burden introduced by the proposed method remains unclear.
- While the overall concept is clear, specific implementation details remain vague, making it difficult to follow.

---

> ### Author Rebuttal · Authors · 2026-03-31
>
> We thank the reviewer for their feedback and for raising important points regarding baselines, computational costs, and implementation clarity. We address each point below.
>
> ## W1: Baseline Strength and Evaluation Depth
> When selecting comparative baselines, it is important to note that abstract latent goal representations are not common in the goal-conditioned RL literature, as previous studies have focused on single observation goals and not abstract goals as suggested in this paper. Hence, we chose goal representation approaches that are conducive to this kind of abstract representation and adapted them accordingly. With this caveat in mind, we have added three more recent baselines, namely DGR [1], TRA [2], and VIP [3], whose results and discussion are provided in the reply to reviewer ZUbk. Regarding evaluation depth, we are of course open to further suggestions. Evaluating pretrained representations on downstream tasks with reward signals is standard practice; however, meaningfully illustrating abstract representations is a non-standard challenge, since the goal-conditioned literature is typically focused on single-observation goals rather than the abstract goals proposed in our work.
>
> ## W2: Computational Overhead
> Because the compared algorithms differ significantly in design, a precise quantitative comparison of computational cost is not straightforward. For a qualitative assessment, two components in our method introduce additional overhead relative to single-observation baselines. First, sequence or set encoding instead of single-observation encoding, which can be implemented via a GRU/LSTM or a transformer-style set encoder. Second, abstract goal generation via gradient optimization: generating an abstract goal from a single concrete candidate requires $n = 5$ gradient steps (Equation 1), and generating abstract goals from a set of concrete candidates additionally requires one gradient ascent step per candidate followed by selection of the top-$k$ with the smallest gradient norm and then 5 further gradient steps of Equation 3. We will add a short paragraph discussing these computational costs to the discussion section of the paper.
>
> ## W3:  Implementation Clarity
> We will revise the main text to present the architectural details more clearly. Full implementation details are provided in the Appendix and supplemented by code included in the supplementary material.
>
> - [1] Park et al., 2026 Dual Goal Representations
> - [2] Myers et al., 2025 Temporal Representation Alignment: Successor Features Enable Emergent Compositionality in Robot Instruction Following
> - [3] Ma et al., 2023 VIP: Towards Universal Visual Reward and Representation via Value-Implicit Pre-Training

---

> > ### Author Rebuttal · Reviewer_EHQw · 2026-04-03
> >
> > My concerns have been well addressed, especially regarding the strength and choice of baselines.
> > Due to the potentially novel and interesting nature of the idea, I will adjust my score accordingly.
> > I would also encourage the authors to present the implementation details of the method more clearly in the camera-ready version, and to further clarify how it compares to the baselines in terms of computational cost and inference time.

---

### Official Review · Reviewer_tenS · 2026-03-08

**Soundness:** 2
**Presentation:** 3
**Significance:** 3
**Originality:** 3
**Overall Recommendation:** 4
**Confidence:** 3

**Summary:**

This paper studies goal representation learning for self-supervised / unsupervised goal-conditioned RL. The main idea is to learn a hierarchical latent goal space in which more concrete and more abstract goals coexist, and are related by a learned subset-like partial order. Concretely, the paper encodes observation sequences into discrete latent goals and trains an asymmetric energy function to predict whether one sequence is a subsequence / subset-like refinement of another. This learned relation is then used to traverse the latent space toward more abstract or more concrete goals, and to support “abstract hindsight relabeling” during contrastive RL training. The method is evaluated on GridWorld, MemoryMaze, and FetchPush, with adaptation to novel reward functions after pretraining.

**Compliance With Llm Reviewing Policy:**

Affirmed.

**Final Justification:**

The rebuttal addresses several of my concerns constructively. In particular, the clarification that the learned representation is intended to capture set-based subset structure rather than strict temporal order helps resolve my concern about the sequential encoder, and I appreciate the authors’ willingness to moderate the broader abstraction claims. The additional discussion of structural diagnostics is also helpful, although I still view the learned relation as closer to a soft asymmetric abstraction relation than a fully enforced partial-order / lattice structure. My remaining uncertainty mainly concerns the empirical case: if the newly added stronger baselines indeed show that the method remains competitive, that would materially strengthen my assessment. Given the rebuttal, I am inclined to soften my original weak reject toward a more borderline evaluation.

**Key Questions For Authors:**

1. Since the main contribution is a new goal representation, could the authors compare against stronger and more recent latent-goal / unsupervised RL baselines beyond raw observational goals and β-VAE?
A stronger empirical comparison could materially affect my evaluation, since the current baseline set is one of the main reasons I view the empirical support as limited.

2. The paper is motivated in part by temporal proximity and trajectory structure, but Appendix B.1 suggests that shuffled, non-sequential data performs similarly. Why is a sequential encoder still the right architectural choice, and would a permutation-invariant set encoder perform similarly?
A convincing clarification here would help me better understand the true source of the gains and the appropriate scope of the paper’s claims.

3. The paper describes the learned structure using terms such as partial order and lattice, but the learning objective does not explicitly enforce transitivity or antisymmetry. Can the authors quantify order consistency more directly, e.g., via transitivity violation rates, cycle rates, or other structural diagnostics?
A stronger structural analysis here could improve my assessment of the method’s soundness, since the gap between the conceptual framing and the enforced structure is one of my main concerns.

**Limitations:**

yes

**Strengths And Weaknesses:**

Strength:
- The problem is important and well motivated. The paper addresses a genuine limitation of observation-based goals: depending on the observation modality, they can be either overly specific or too ambiguous to support robust generalization. The framing of abstraction as a subset-like containment relation is elegant, and the idea of placing both concrete and abstract goals in a shared latent space is conceptually appealing.
- The method is coherent end-to-end. The overall pipeline — representation learning → asymmetric energy modeling → abstract relabeling → downstream adaptation to novel rewards — is well connected and easy to follow. This gives the paper a clear methodological story rather than a collection of loosely related components.

Weakness:
- The empirical evaluation is encouraging, but in my view still somewhat limited for ICML. In particular, the baseline set mainly consists of raw observational goals and a β-VAE baseline. Since the core contribution is about goal representation learning, I would have liked to see stronger and more recent comparisons from the latent-goal and unsupervised RL literature. The paper would also be strengthened by validation on a broader or more standard set of RL benchmarks.
- The conceptual framing is appealing, but some of the structural claims seem somewhat stronger than what the method explicitly enforces. The paper uses the language of learned partial orders, hierarchies, and lattices, yet the learning objective does not directly impose key order-theoretic properties such as transitivity, antisymmetry, or consistency of join and meet operations. As a result, the method currently appears closer to learning a soft asymmetric abstraction relation than a fully structured hierarchy in the formal sense.
-The scope of the abstraction claim may be narrower than the paper suggests. The method is primarily validated on spatial / coverage-like abstractions induced by containment relations between trajectories or sets of observations. This is a meaningful and useful setting, but it is only one kind of abstraction. At present, the evidence does not yet support a broader claim that the method is a general solution for hierarchical goal abstraction in RL.

---

> ### Author Rebuttal · Authors · 2026-03-31
>
> We thank the reviewer for their insightful review and for the careful engagement with both the conceptual framing and the empirical evaluation.
>
> ## W1/Q1:  Stronger and More Recent Baselines
>
> When selecting comparative baselines, it is important to note that abstract latent goal representations are not common in the goal-conditioned RL literature, as previous studies have focused on single observation goals and not abstract goals as suggested in this paper. Hence, we chose goal representation approaches that are conducive to this kind of abstract representation and adapted them accordingly. With this caveat in mind, we have added three more recent baselines, namely DGR [1], TRA [2], and VIP [3], whose results and discussion are provided in the reply to reviewer ZUbk.
>
> ## Q2: Sequential Encoder vs. Permutation-Invariant Set Encoder
>
> We conducted the shuffled-sequence experiment in the Appendix specifically to demonstrate that our learning objective extracts set-based properties compatible with the imposed subset relation and does not rely on the actual sequential ordering of trajectories. We chose a GRU-based RNN for its simplicity and efficiency, but we agree that a permutation-invariant set encoder would be a principled and suitable replacement, and we will note this as a design alternative in the paper.
>
> ## W2/Q3: Order-Theoretic Properties and Structural Analysis
>
> We view our approach as learning a soft partial order. While we do not explicitly enforce transitivity, our training objective does impose constraints on anti-symmetry (through reversed samples as negative examples), join operations (through concatenation of sets of trajectories), and bounds (via the most concrete and most abstract goals that bound the goal space). In practice we observe that transitivity holds to a large extent, as shown in Supplementary Figure 12, which illustrates repeated goal abstraction traversals and confirms that transitivity is maintained across steps. Simply optimizing the energy function yields one possible trajectory of abstract goals, but we also observe that this approach does not always find, for example, compositional goals.
>
> To further analyze the structure of the latent space, we conducted additional experiments (to be added to the paper) based on combining goals that are easiest to join, as measured by the energy gradient magnitude of one goal with respect to another when optimizing Equation 3. Goals with small mutual gradients share spatial or observational features. With repeated application of this mechanism, spatial abstraction increases (to at most approximately 20% of the environment's states) while transitivity (measured by the energy) remains high over 10 abstraction steps, and successive goals cover on average 70% of the preceding goals (consistent with the analysis in Supplementary Figure 12). We will include a more in-depth structural analysis in the final version.
>
> ## W3: Scope of the Abstraction Claim
> We agree that our method explicitly focuses on spatial abstraction, and we will moderate the corresponding claims in the paper. That said, we believe the subset interpretation is valid for abstraction more broadly: while our current instantiation exploits temporal containment, other notions of similarity could in principle be used to define the containment relation for different kinds of abstraction. We will make this distinction clearer in the revised manuscript.
>
> - [1] Park et al., 2026 Dual Goal Representations
> - [2] Myers et al., 2025 Temporal Representation Alignment: Successor Features Enable Emergent Compositionality in Robot Instruction Following
> - [3] Ma et al., 2023 VIP: Towards Universal Visual Reward and Representation via Value-Implicit Pre-Training

---

> > ### Author Rebuttal · Reviewer_tenS · 2026-04-03
> >
> > My concerns have been adequately addressed.

---

### Official Review · Reviewer_Dnr2 · 2026-03-11

**Soundness:** 3
**Presentation:** 3
**Significance:** 3
**Originality:** 4
**Overall Recommendation:** 6
**Confidence:** 4

**Summary:**

This paper explores new methods to represent goals in reinforcement learning, using encodings that support partial orderings of goals from more concrete to more abstract.  Goals are represented by fixed-width encodings of variable-length observation sequences (to address partial observability).  The partial ordering of goals is determined based on the subsequence relationship of the encoded observation sequences, since a subsequence will tend to correspond to a smaller portion of the state space and hence a more specific/concrete goal.

The goal encoder networks, and a soft function approximator for the partial-ordering indicator, are trained jointly with a contrastive learning approach.  After training, with the partial-order approximator fixed, one can numerically optimize its inputs to traverse the abstraction hierarchy, or to find goal encodings that are well-aligned with previously unseen reward functions.  Those goals can then be used to drive a goal-conditioned policy towards high returns.

The method is empirically tested on three diverse environments that cover discrete and continuous state/action spaces and observation modalities such as LIDAR and vision.  The results show that abstract goal encodings can better capture certain reward functions than the very concrete (single-observation) goal representations commonly used in goal-conditioned RL.  The method also compares favorably with two baselines.

**Compliance With Llm Reviewing Policy:**

Affirmed.

**Final Justification:**

The most significant issue raised in my review - namely, comparison with strong baselines - was addressed in the rebuttal.  Hence I increased my rating by one point.

**Key Questions For Authors:**

Responses to any/all of the weaknesses raised above would be appreciated.

**Limitations:**

Yes

**Strengths And Weaknesses:**

Strengths:

Goal abstraction is an important area of RL research, and to my knowledge this paper approaches it in a rather original way.  The paper does not retrace the beaten paths of most RL work, and is more than an incremental adjustment to previous methods.  The writing is clear and the methodology is well-reasoned.  In terms of significance, the method is not necessarily state-of-the-art on each task where it is tested, but it does provide new insights and ideas, demonstrating their feasibility and opening interesting avenues for future work.

Weaknesses:

- It is not so clear if the selected baselines are SoTA for the selected tasks.  It would strengthen the paper to also compare the proposed method with current SoTA, even if beating SoTA is not the primary objective of this work.
- In section 3, it was unclear why the energy function used as similarity, when earlier it was supposed to indicate the partial ordering.
- The goal encoder architectures were described, but the learned energy function architecture was less clear.  I recommend adding more detail about this on page 3.
- In equation 3, it was not clear how binary cross entropy is justified unless rewards are sparse, binary 0|1 signals.  Are they? Later on this seems to be the case, but would be best to state it here explicitly.
- Figure 7 indicates that observational rewards work better in the MemoryMaze task.  Page 6 (novel rewards section) would benefit from more discussion of why this might have occurred.
- The paper abstract mentions two issues with single-observation goal representations: either too concrete, or too abstract (due to partial observability).  However, as far as I can tell, the method only addresses the first issue, since length-1 observation sequences are treated as most concrete despite the partial observability.

Minor suggestions:

- I would recommend some different word choice that, in my mind, would enhance the clarity of the paper.  In particular, "subsequence" might be more specific and accurate than "subset".  Also, I usually think of an energy function as a function of a single state (not a pair), so maybe "characteristic" function is better without reference to the energy concept.
- Use backticks (``) for open quotes in latex (line 125 left)
- In the fetchpush description, there is a missing year in the Eysenbach citation

---

> ### Author Rebuttal · Authors · 2026-03-31
>
> We sincerely thank the reviewer for their positive assessment of the paper's originality. We address each point below.
>
> ## W1:  Baseline Strength
> When selecting comparative baselines, it is important to note that abstract latent goal representations are not common in the goal-conditioned RL literature, as previous studies have focused on single observation goals and not abstract goals as suggested in this paper. Hence, we chose goal representation approaches that are conducive to this kind of abstract representation and adapted them accordingly. With this caveat in mind, we have added three more recent baselines, namely DGR [1], TRA [2], and VIP [3], whose results and discussion are provided in the reply to Reviewer ZUbk.
>
> ## W2: Energy Function as Similarity
>
> We agree that similarity may be a misleading term here and will clarify this in the revised manuscript. The intended meaning is a compatibility measure in latent space: we use the energy function to check whether an abstract goal respects the subset relation with respect to a set of concrete candidate goals. By contrast, the β-VAE baseline measures compatibility via Euclidean distance, capturing how a latent abstract representation relates to its corresponding concrete candidates. We will revise the wording accordingly.
>
> ## W3:  Energy Function Architecture
> The energy function is a simple MLP that takes the two goal inputs concatenated as input. We will add this detail explicitly to the main text.
>
> ## W4:  Binary Cross-Entropy Justification
> For all evaluation tasks the rewards are indeed binary. We will state this explicitly in the manuscript at the point where Equation 3 is introduced.
>
> ## W5:  Observational Rewards in MemoryMaze
> Thank you for pointing this out. Observational rewards in MemoryMaze (e.g., observing a red wall) are perfectly captured by a single raw image, making abstract goal optimization unnecessary. Furthermore, learning abstract spatial goals introduces a unique challenge: different ego-centric views of the exact same spatial region yield completely different visual observations. While our goal representation successfully encodes spatial rewards, the downstream policy may still be heavily dominated by raw color information. We will add this discussion to Section 3 of the paper.
>
> ## W6:  Partial Observability and Abstraction
> We appreciate this observation, as it highlights a nuanced aspect of our representation space. In our framework, abstraction is defined by coverage: a larger set of represented states corresponds to a more abstract goal. Under partial observability, a single ambiguous observation (e.g., a yellow corner) does not represent a single concrete state; it is highly aliased and effectively encodes a broad set of states sharing that visual appearance. Treating such an observation as "most concrete" therefore fails. Our soft partial-order objective naturally accommodates this: encoding a single corner image results in an abstract "any corner" goal, while encoding a sequence of multiple observations provides the temporal and spatial context needed to disambiguate the location, yielding a highly specific "this exact corner" goal. In this sense, a longer sequence can represent fewer possible states, making the goal more concrete.
>
> ## Minor suggestions
>
> Regarding terminology, we use energy function to align with the vocabulary used in the Energy-Based Model (EBM) [4] literature, where such functions commonly take multiple inputs to measure compatibility. We appreciate the suggestion of "characteristic function" as an alternative and will note this connection in the text. We will also fix the LaTeX open quotes and add the missing year to the Eysenbach citation.
>
> * [1] Park et al., 2026 Dual Goal Representations
> * [2] Myers et al., 2025 Temporal Representation Alignment: Successor Features Enable Emergent Compositionality in Robot Instruction Following
> * [3] Ma et al., 2023 VIP: Towards Universal Visual Reward and Representation via Value-Implicit Pre-Training
> * [4] LeCun et al., 2006 A tutorial on energy-based learning.

---

> > ### Author Rebuttal · Reviewer_Dnr2 · 2026-04-02
> >
> > The rebuttal addresses all my concerns. In particular, the inclusion of stronger baselines is a substantial improvement, and I am adjusting my rating accordingly.

---

### Official Review · Reviewer_ZUbk · 2026-03-13

**Soundness:** 3
**Presentation:** 3
**Significance:** 3
**Originality:** 3
**Overall Recommendation:** 4
**Confidence:** 4

**Summary:**

The authors propose an approach to construct hierarchical latent goal spaces that integrate both concrete and abstract goals using an energy function based approach. In this space, the approach is able to learn a both concrete and abstract subgoals, thus forming a hierarchy of subgoals using temporal contrastive learning and sequence-level hindsight relabeling. In navigation and robotic manipulation tasks, they demonstrate better generalization to novel tasks versus observational goals.

**Compliance With Llm Reviewing Policy:**

Affirmed.

**Final Justification:**

The rebuttal addressed my concerns, and I will keep a positive score.

**Key Questions For Authors:**

1. Can the authors provide evaluations on harder tasks (e.g. OGBench)?
2. Can the authors add stronger GCRL or HRL baselines?

**Limitations:**

Please see weaknesses above.

**Strengths And Weaknesses:**

Strengths:
1. The approach uses an asymmetric energy function to learn a partial order between sequence observations, thus inducing a hierarchy from concrete to abstract goals.
2.  The framework successfully allows pre-trained agents to adapt to previously unseen reward functions simply by finding the best-fitting goal in the latent space.
3. The empirical results show that their approach significantly outperforms the beta-VAE and observational baselines.

Weaknesses:
1. The empirical comparisons could be stronger. The paper only compares against a beta-VAE and raw observational goals. Comparing against more recent state-of-the-art goal-conditioned RL or hierarchical RL methods (e.g. Director [1]) would make the results much more convincing.
2. The evaluation is restricted to relatively simple environments (GridWorld, MemoryMaze, and FetchPush). Evaluating the approach on more complex, high-dimensional continuous control tasks or real-world robotic environments would better prove the method's scalability.

[1] Deep Hierarchical Planning from Pixels, Hafner et al, 2022.

---

> ### Author Rebuttal · Authors · 2026-03-31
>
> We thank the reviewer for their constructive feedback and the suggestions to strengthen our empirical evaluation.
> ## W1/Q2:  Baselines
> When selecting comparative baselines, it is important to note that abstract latent goal representations are not common in the goal-conditioned RL literature, as previous studies have focused on single observation goals and not abstract goals as suggested in this paper. Hence, we chose goal representation approaches that are conducive to this kind of abstract representation and adapted them accordingly. With this caveat in mind, we have added three more recent goal-representation learning baselines from ICLR 2026, NeurIPS 2025, and ICLR 2023: DGR [1], TRA [2], and VIP [3]. All three allow pre-training of the representation and exploit temporal similarity:
> - DGR represents goals as distances to all possible states. Similarity is defined as  $V(s,g) = \psi(s)^\top\varphi(g)$.
> - TRA parametrizes similarity as DGR but is trained with an InfoNCE loss.
> - VIP uses an embedding learned from temporal distance: $V(s,g) = -\|\varphi(s)-\varphi(g)\|_2$, trained contrastively
>
> All three approaches were originally designed to encode single observations as goals; however, we use $\varphi$ as goal representation and optimize it directly using the respective similarity measure to encode reward functions. Remarkably, all three methods can represent abstract goals in latent space even though the original authors never explored this capability, suggesting that temporal contrastive objectives implicitly induce some hierarchical structure. However, none of them provide a principled mechanism for hierarchy traversal, which our method explicitly does via the energy function.
>
> ### GridWorld
> Model|Obs. Goal Att.|Obs. Total Rew.|Spatial Goal Att.|Spatial Total Rew.
> -|-|-|-|-
> Ours (relabel only)|0.57±0.20|0.43±0.21|0.52±0.06|0.56±0.05
> Ours+Eq.1*|0.58±0.20|0.34±0.22|0.57±0.09|0.66±0.10
> Ours+Eq.3**|**0.68±0.17**|**0.48±0.20**|**0.65±0.11**|**0.76±0.12**
> β-VAE|0.58±0.22|0.43±0.24|0.52±0.10|0.56±0.11
> VIP|0.47±0.27|0.36±0.25|0.52±0.23|0.57±0.29
> TRA|0.51±0.23|0.38±0.25|0.59±0.21|0.65±0.24
> DGR|0.54±0.23|0.33±0.16|0.60±0.16|0.70±0.20
> Observation|0.58±0.20|0.31±0.19|0.38±0.16|0.29±0.24
>
> ### MemoryMaze
> Model|Obs. Goal Att.|Obs. Total Rew.|Spatial Goal Att.|Spatial Total Rew.
> -|-|-|-|-
> Ours (rel.)|**0.57±0.22**|0.44±0.31|**0.27±0.15**|**0.30±0.16**
> Ours+Eq.1*|0.52±0.25|0.41±0.31|0.23±0.14|0.26±0.16
> Ours+Eq.3**|0.56±0.21|0.43±0.29|0.25±0.15|0.27±0.15
> β-VAE|0.50±0.27|0.42±0.33|0.25±0.16|0.27±0.18
> VIP|0.47±0.26|0.35±0.31|0.22±0.11|0.22±0.10
> TRA|0.51±0.24|0.40±0.31|0.22±0.14|0.25±0.16
> DGR|0.54±0.25|**0.45±0.31**|0.26±0.19|0.30±0.21
> Observation|0.45±0.29|0.32±0.34|0.19±0.13|0.19±0.15
>
> ### FetchPush
> Model|Grip. Goal Att.|Grip. Total Rew.|Obj. Goal Att.|Obj. Total Rew.
> -|-|-|-|-
> Ours (rel.)|0.59±0.35|0.36±0.25|0.55±0.29|0.40±0.22
> Ours+Eq.1*|0.61±0.32|0.37±0.23|0.59±0.27|0.55±0.25
> Ours+Eq.3**|0.70±0.34|**0.58±0.31**|**0.71±0.32**|**0.60±0.33**
> β-VAE|0.55±0.33|0.36±0.24|0.44±0.36|0.38±0.32
> VIP|**0.71±0.36**|0.56±0.36|0.71±0.28|0.55±0.28
> TRA|0.62±0.35|0.45±0.28|0.56±0.35|0.43±0.32
> DGR|0.56±0.34|0.41±0.24|0.56±0.33|0.44±0.28
> Observation|0.50±0.28|0.27±0.18|0.28±0.32|0.19±0.17
>
> (*) Eq.1 to find more abstract goals starting from a relabeled goal
>
> (**) Eq.3 with rewards defined by gradient magnitude to candidate goals (see reply to reviewer tenS)
>
> ## W1/Q1:  Director
> Our paper focuses strictly on model-free approaches where the goal representation is agnostic to the underlying GCRL algorithm. Director is a model-based, hierarchical method that learns goals over hidden states, so a direct empirical comparison falls outside our model-free scope. That said, integrating our approach with model-based RL is a highly promising direction for future work.
>
> ## W2/Q1:  Complex Environments
> While real-world robotic evaluation is an important next step, many complex benchmarks use offline RL or imitation learning with preexisting data, sidestepping exploration. We focus on online RL, where exploration and abstract goal discovery are co-dependent, a challenge absent from the offline setting.
>
> We note that our evaluation suite is challenging on its own terms:
> - **FetchPush**: image-based (harder than standard state-based).
> - **GridWorld & MemoryMaze**: ego-centric observations (LiDAR and images), imposing partial observability.
>
> All tasks require overcoming partial observability and learning from complex sensory inputs. There is also a fundamental methodological mismatch: our analysis evaluates performance on novel reward functions representing abstract goals, whereas OGBench defines goals strictly as concrete target observations or states.
>
> - [1] Park et al., 2026 Dual Goal Representations
> - [2] Myers et al., 2025 Temporal Representation Alignment: Successor Features Enable Emergent Compositionality in Robot Instruction Following
> - [3] Ma et al., 2023 VIP: Towards Universal Visual Reward and Representation via Value-Implicit Pre-Training

---

> > ### Author Rebuttal · Reviewer_ZUbk · 2026-04-04
> >
> > I thank the authors for their responses. My concerns are sufficiently addressed, and I will maintain my positive score.

---

### Decision · Program_Chairs · 2026-04-30

**Decision:**

Accept (regular)

**Comment:**

This paper presents a method for self-supervised goal-conditioned reinforcement learning (GCRL). The authors propose to construct a latent goal space from pre-collected trajectories. Specifically, the method encodes trajectories into latent embeddings and learns a subset relation between them using an energy function. These learned goals further allow the RL agent to plan across different levels of abstraction and provide high-level abstract goals. Experiment results on navigation tasks and robotic control tasks show that the proposed method can better capture abstract reward functions and lead to improved performance.

All reviewers agree that this paper is well-motivated and well-written, and it addresses an important problem in GCRL. The idea of learning abstraction embeddings through a subset relationship and constructing hierarchical goal spaces is novel. The overall RL training pipeline is also well-connected. Empirical results look promising and demonstrate the potential of the proposed method. Although some reviewers question whether the method is SOTA, the authors added additional baselines in their rebuttal, which I think should be included in the revision. Overall, I believe this is good work and therefore recommend acceptance.